# Systematic analysis of naturally occurring insertions and deletions that alter transcription factor spacing identifies tolerant and sensitive transcription factor pairs

Zeyang Shen[1,2], Rick Z Li[1], Thomas A Prohaska[3], Marten A Hoeksema[1,4†], Nathan J Spann[1], Jenhan Tao[1], Gregory J Fonseca[1,5‡], Thomas Le[6], Lindsey K Stolze[7], Mashito Sakai[1,8§], Casey E Romanoski[7], Christopher K Glass[1,3*]

[1]Department of Cellular and Molecular Medicine, School of Medicine, University of California San Diego, La Jolla, United States; [2]Department of Bioengineering, Jacobs School of Engineering, University of California San Diego, La Jolla, United States; [3]Department of Medicine, School of Medicine, University of California San Diego, La Jolla, United States; [4]Department of Medical Biochemistry, Experimental Vascular Biology, Amsterdam Infection and Immunity, Amsterdam Cardiovascular Sciences, Amsterdam UMC, University of Amsterdam, Amsterdam, Netherlands; [5]Department of Medicine, McGill University, Montreal, Canada; [6]Division of Biological Sciences, University of California San Diego, La Jolla, United States; [7]Department of Cellular and Molecular Medicine, College of Medicine, University of Arizona, Tucson, United States; [8]Department of Biochemistry and Molecular Biology, Nippon Medical School, Tokyo, Japan

*For correspondence:
ckg@ucsd.edu

Present address: †Department of Medical Biochemistry, Experimental Vascular Biology, Amsterdam Infection and Immunity, Amsterdam Cardiovascular Sciences, Amsterdam UMC, University of Amsterdam, Amsterdam, Netherlands; ‡Department of Medicine, McGill University, Montreal, Canada; §Department of Biochemistry and Molecular Biology, Nippon Medical School, Tokyo, Japan

Competing interest: The authors declare that no competing interests exist.

**Abstract** Regulation of gene expression requires the combinatorial binding of sequence-specific transcription factors (TFs) at promoters and enhancers. Prior studies showed that alterations in the spacing between TF binding sites can influence promoter and enhancer activity. However, the relative importance of TF spacing alterations resulting from naturally occurring insertions and deletions (InDels) has not been systematically analyzed. To address this question, we first characterized the genome-wide spacing relationships of 73 TFs in human K562 cells as determined by ChIP-seq (chromatin immunoprecipitation sequencing). We found a dominant pattern of a relaxed range of spacing between collaborative factors, including 45 TFs exclusively exhibiting relaxed spacing with their binding partners. Next, we exploited millions of InDels provided by genetically diverse mouse strains and human individuals to investigate the effects of altered spacing on TF binding and local histone acetylation. These analyses suggested that spacing alterations resulting from naturally occurring InDels are generally tolerated in comparison to genetic variants directly affecting TF binding sites. To experimentally validate this prediction, we introduced synthetic spacing alterations between PU.1 and C/EBPβ binding sites at six endogenous genomic loci in a macrophage cell line. Remarkably, collaborative binding of PU.1 and C/EBPβ at these locations tolerated changes in spacing ranging from 5 bp increase to >30 bp decrease. Collectively, these findings have implications for understanding mechanisms underlying enhancer selection and for the interpretation of non-coding genetic variation.

## Editor's evaluation

Transcription factors (TFs) bind to the DNA in a sequence-specific manner at TF binding sites (TFBSs) to control gene transcription. Hence, characterizing how TFs interact with DNA is key to uncover how gene regulation occurs and how this process can be disrupted in diseases. While the binding properties of a large portion of human TFs are well characterized, a remaining challenge lies in our knowledge of how TFs interact cooperatively at regulatory elements, either forming dimers or co-binding the same regions. In this manuscript, Shen et al. explored spacing patterns between TFBSs using previously published data sets and revealed that the dominant pattern is a relaxed range of spacing between collaborative factors and tolerance for InDels that change the TFBS spacing.

## Introduction

Genome-wide association studies (GWASs) have identified thousands of genetic variants associated with diseases and other traits (*MacArthur et al., 2017*; *Visscher et al., 2017*). Single nucleotide polymorphisms (SNPs) and short insertions and deletions (InDels) represent common forms of these variants. The majority of GWAS variants fall at non-protein-coding regions of the genome, suggesting their effects on gene regulation (*Farh et al., 2015*; *Ward and Kellis, 2012*). Gene expression is regulated by transcription factors (TFs) in a cell-type-specific manner. A TF can bind to a specific set of short, degenerate DNA sequences at promoters and enhancers, often referred to as TF binding motif. Active promoters and enhancers are selected by combinations of sequence-specific TFs that bind in an inter-dependent manner to closely spaced motifs. SNPs and InDels can create or disrupt TF binding sites by mutating motifs and are a well-established mechanism for altering gene expression and biological function (*Behera et al., 2018*; *Deplancke et al., 2016*; *Grossman et al., 2017*; *Heinz et al., 2013*). InDels can additionally change spacing between TF binding sites, but it remains unknown the extent to which altered spacing is relevant for interpreting genetic variation in human populations or between animal species.

Previous studies reported two major categories of motif spacing between inter-dependent TFs (*Slattery et al., 2014*). One category refers to the enhanceosome model (*Slattery et al., 2014*) that requires specific or 'constrained' spacing. It is mainly provided by TFs that form ternary complexes recognizing composite binding motifs, exemplified by GATA, Ets, and E-box TFs in mouse hematopoietic cells (*Ng et al., 2014*), MyoD and other cell-type-specific factors in muscle cells (*Nandi et al., 2013*), and Sox2 and Oct4 in embryonic stem cells (*Rodda et al., 2005*). In vitro studies of the binding of pairwise combinations of ~100 TFs to a diverse library of DNA sequences identified 315 out of 9400 possible interactive TF pairs that select composite elements with constrained positions of the respective recognition motifs (*Jolma et al., 2015*). Constrained spacing required for the optimal binding and function of interacting TFs can also occur between independent motifs, such as occurs at the interferon-β enhanceosome (*Panne, 2008*). In comparison to constrained spacing, another category of motif spacing allows TFs to interact over a relatively broad range (e.g., 100–200 bp), which we call 'relaxed' spacing and is equivalent to the billboard model (*Slattery et al., 2014*). This type of spacing relationship is observed in collaborative or co-occupied TFs that do not target promoters or enhancers as a ternary complex (*Heinz et al., 2010*; *Jiang and Singh, 2014*; *Sönmezer et al., 2021*).

Substantial evidence indicates that the two categories of spacing requirement can experience a different level of impact from genetic variation. Reporter assays examining synthetic alterations of motif spacing revealed examples of TFs that require constrained spacing and have high sensitivity of TF binding and gene expression on spacing (*Farley et al., 2015*; *Ng et al., 2014*; *Panne, 2008*). On the contrary, flexibility in motif spacing has been demonstrated using reporter assays in *Drosophila* (*Menoret et al., 2013*) and HepG2 cells (*Smith et al., 2013*). However, these studies did not distinguish the impact of altered spacing on TF binding or subsequent recruitment of co-activators required for gene activation. Moreover, it remains unknown the extent to which these findings are relevant to spacing alterations resulting from naturally occurring genetic variation.

To investigate the effects of altered spacing on TF binding and function, we first characterized the genome-wide binding patterns of 73 TFs based on their binding sites determined by chromatin immunoprecipitation sequencing (ChIP-seq). We developed a computational framework that assigned each spacing relationship to 'constrained' or 'relaxed' category and associated spacings to the naturally

occurring InDels observed in human populations to study the selective constraints of different spacing relationships. As specific case studies, we leveraged natural genetic variation in numerous human samples and from five strains of mice to study the effect size of spacing alterations on TF binding activity and local histone acetylation. These findings suggested that InDels altering spacing are generally less constrained and well tolerated when they occur between TF pairs with relaxed spacing relationships. Finally, we experimentally validated substantial tolerance in spacing for macrophage lineage-determining TFs (LDTFs), PU.1 and C/EBPβ, by introducing a wide range of InDels between their respective binding sites at representative endogenous genomic loci using CRISPR/Cas9 mutagenesis in mouse macrophages.

## Results

### TFs primarily co-bind with relaxed spacing

We characterized spacing relationships for 73 TFs of K562 cells covering diverse TF families (*Hu et al., 2019*) based on the ChIP-seq data from ENCODE data portal (*Davis et al., 2018*). After obtaining reproducible ChIP-seq peaks, we used the corresponding position weight matrix (PWM) of each TF (*Supplementary file 1*) to identify the locations of high-affinity binding sites that are less than 50 bp from peak centers (*Figure 1A*; *Figure 1—figure supplement 1*); 42% of peaks on average contain at least one binding site of corresponding TF (*Supplementary file 1*). The peaks of every pair of TFs were then merged, and at the overlapping peaks indicating co-binding events, the edge-to-edge spacings were calculated between TF binding sites and then aggregated to show a distribution within ±100 bp. To categorize spacing relationships, we used Monte Carlo procedures to obtain an empirical p-value to find significant spacing constraints and used Kolmogorov–Smirnov (KS) test to test for a relaxed spacing relationship against random distribution.

We applied this computational framework to all possible pairs of TFs. By dissecting each TF's binding sites based on their spacing relationships with co-binding TFs, we found that 45 of the 73 TFs examined exclusively exhibited relaxed spacing relationships with other TFs (*Figure 1B*). Twenty-five factors could participate in either relaxed or constrained interactions, depending on the specific co-binding TFs. Only three TFs interacted with only constrained spacing, some of which might show additional relaxed spacing relationships by expanding the current set of TFs. The significant pairwise patterns of relaxed and constrained spacing relationships are illustrated in *Figure 1C*. Among 29 TF pairs with constrained spacing relationships, most bind closely to each other within 15 bp spacing (*Figure 1—figure supplement 2*). Some of these TF pairs have been reported to recognize composite motifs such as GATA1-TAL1 and NFATC3-FOSL1 (*Macián et al., 2001*; *Ng et al., 2014*; *Figure 1D*; *Figure 1—figure supplement 3*), and some are novel constrained spacing patterns discovered by our analysis such as MEF2A-JUND and CEBPB-TEAD4 (*Figure 1—figure supplement 3*). TFs exhibiting relaxed spacing are exemplified by ETV1-TAL1 and JUND-KLF16, in which the frequency of co-binding progressively declines with distance from the center of the reference TF (*Figure 1D*). We also saw frequent relaxed spacing between TFs in the same family. For instance, despite the similar motifs recognized by AP-1 factors, many of these TFs were found to co-localize at non-overlapping nearby positions. In addition, the same type of spacing relationship is usually observed in different motif orientations (*Figure 1D*), consistent with previous findings (*Lis and Walther, 2016*).

We downloaded the ChIP-seq data of HepG2 cells from ENCODE and processed them with the same pipeline as for K562 cells. The same TF pairs can have similar spacing relationships in different cell types, exemplified by CEBPB and JUND in K562 and HepG2 cells (*Figure 1—figure supplement 4*). Despite more frequent binding events occurring at specific spacings for constrained TF pairs or at closer spacings for relaxed TF pairs, the binding activities quantified by ChIP-seq tags were indifferent at various spacings, suggesting that the spacing preference is not a determinant of TF binding activity (*Figure 1—figure supplement 5*).

Since DNA repetitive regions such as transposable elements are known to harbor TF binding sites and specific TF co-binding (*Bourque et al., 2008*; *Kunarso et al., 2010*), we further examined whether the spacing relationships of TFs could be different in repetitive and nonrepetitive regions. To study this, we applied the same pipeline to the subsets of TF ChIP-seq peaks in repetitive and nonrepetitive regions. As a result, most of the relaxed spacing relationships remained regardless of repetitive or nonrepetitive regions (*Figure 1—figure supplement 6*). Some constrained TF pairs, however, showed

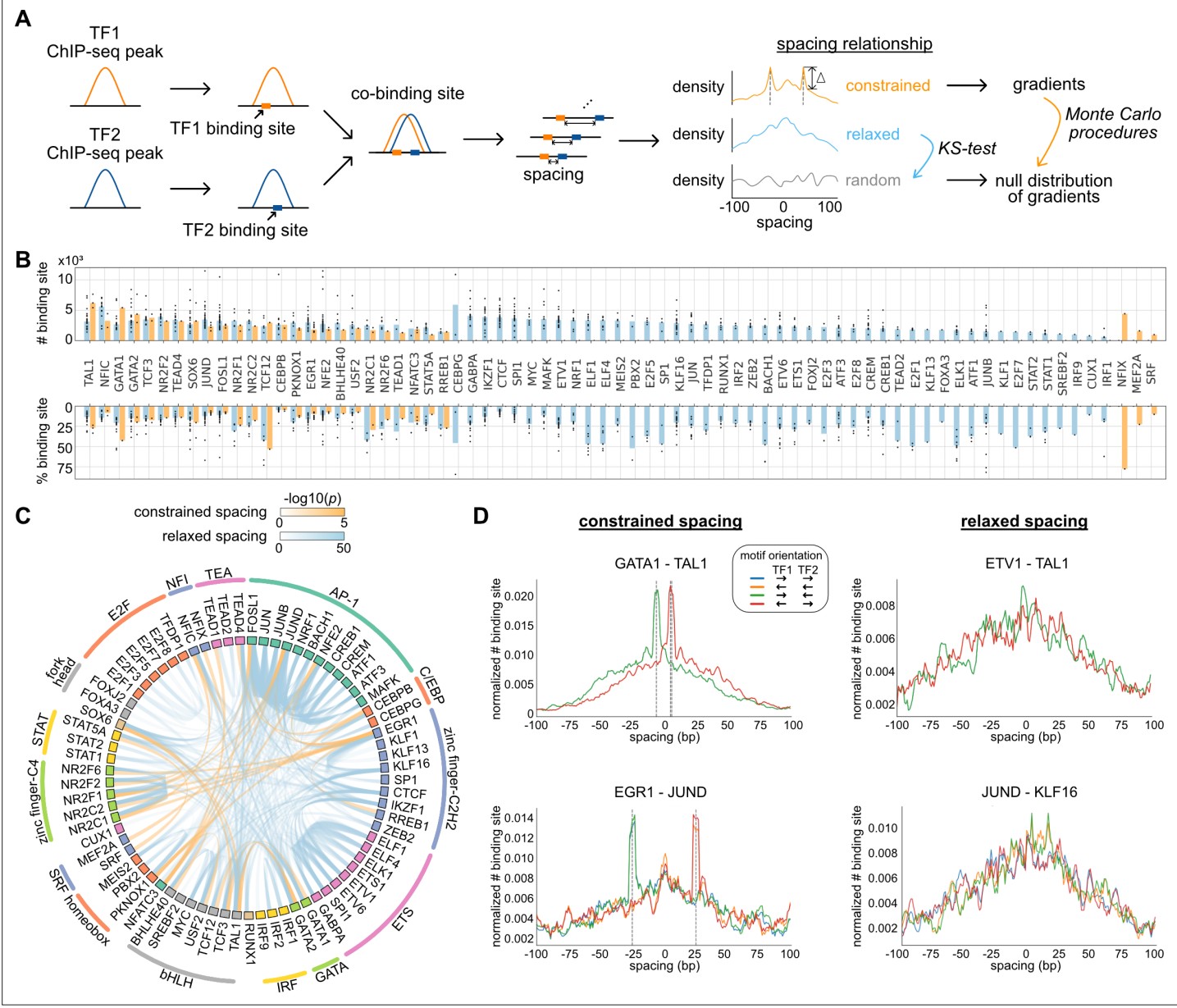

**Figure 1.** Characterization of spacing relationships for transcription factor (TF) pairs. (**A**) Schematic of data analysis pipeline for characterizing the spacing relationships based on TF chromatin immunoprecipitation sequencing (ChIP-seq) data. (**B**) Dissection of TF binding sites for TFs in K562 cells based on spacing relationships with co-binding TFs. Each dot represents a TF pair. The bar heights indicate medians. (**C**) Circos plot summarizing spacing relationships for all the TF pairs analyzed. Orange and blue bands represent significant constrained and relaxed spacing relationships, respectively. Color opacity indicates the level of significance. TFs are grouped and colored by TF family. (**D**) The spacing distributions of example TF pairs with constrained spacing or relaxed spacing relationships. Dashed lines indicate the significant constrained spacings. Since TAL1 motif is completely palindromic, the motif orientation is only differentiated by its co-binding partners.

The online version of this article includes the following source data and figure supplement(s) for figure 1:

**Source data 1.** The numbers of co-binding sites for every pair of 73 transcription factors (TFs).

**Source data 2.** Statistical test results for significant transcription factors (TF) pairs.

**Figure supplement 1.** Effects of different motif scanning criteria.

**Figure supplement 2.** Constrained spacings for the significant transcription factor (TF) pairs with constrained spacing relationships.

**Figure supplement 3.** Examples of transcription factor (TF) pairs with constrained spacing relationships.

**Figure supplement 4.** Comparison of the spacing relationships of same transcription factor (TF) pairs in different cell types.

**Figure supplement 5.** Transcription factor (TF) chromatin immunoprecipitation sequencing (ChIP-seq) tag counts versus spacing for representative TF

*Figure 1 continued on next page*

*Figure 1 continued*

pairs in *Figure 1D*.

**Figure supplement 6.** Comparison between all peaks and peaks only at nonrepetitive regions based on their Kolmogorov–Smirnov (KS) test p-values used to test for relaxed spacing relationship.

**Figure supplement 7.** The spacing relationship of EGR1-JUND based on all co-binding peaks (right) or peaks at repetitive regions, specifically SINEs (left).

constrained spacing only in repetitive regions and not in nonrepetitive regions (*Figure 1—source data 2*). For example, EGR1 and JUND exhibited a constrained spacing at 29 bp (*Figure 1D*), but this relationship is observed specifically in SINEs (*Figure 1—figure supplement 7*). Such observation is consistent with previous studies that discovered specific motif pairs in repetitive regions (*Wang et al., 2012*).

## Natural genetic variants altering spacing between relaxed TFs are associated with less deleteriousness in human populations

Based on a global view of the TF spacing relationships, we then studied whether these relationships associate with different levels of sensitivity to spacing alterations. Here, we leveraged more than 60 million InDels from gnomAD data (*Karczewski et al., 2020*), which were based on more than 75,000 genomes from unrelated individuals. We mapped these InDels to the TF binding sites of representative TF pairs with constrained and relaxed spacing identified in K562 cells. We found that InDels between TF binding sites have similar sizes compared to those at binding sites and those in background regions, the majority of which are less than 5 bp (*Figure 2A*). Next, we divided these InDels based on their allele frequency (AF) and allele count (AC) into high-frequency variants (AF>0.01%), rare variants (AF<0.01%, AC>1), and singletons (AC = 1). Most of the InDels are singletons or rare variants (*Figure 2—figure supplement 1*; *Figure 2—source data 1*). We compared the enrichment of different categories of InDels between or at TF binding sites (*Figure 2B*; *Figure 2—figure supplement 2*). The InDel compositions at TF binding sites were not significantly different between

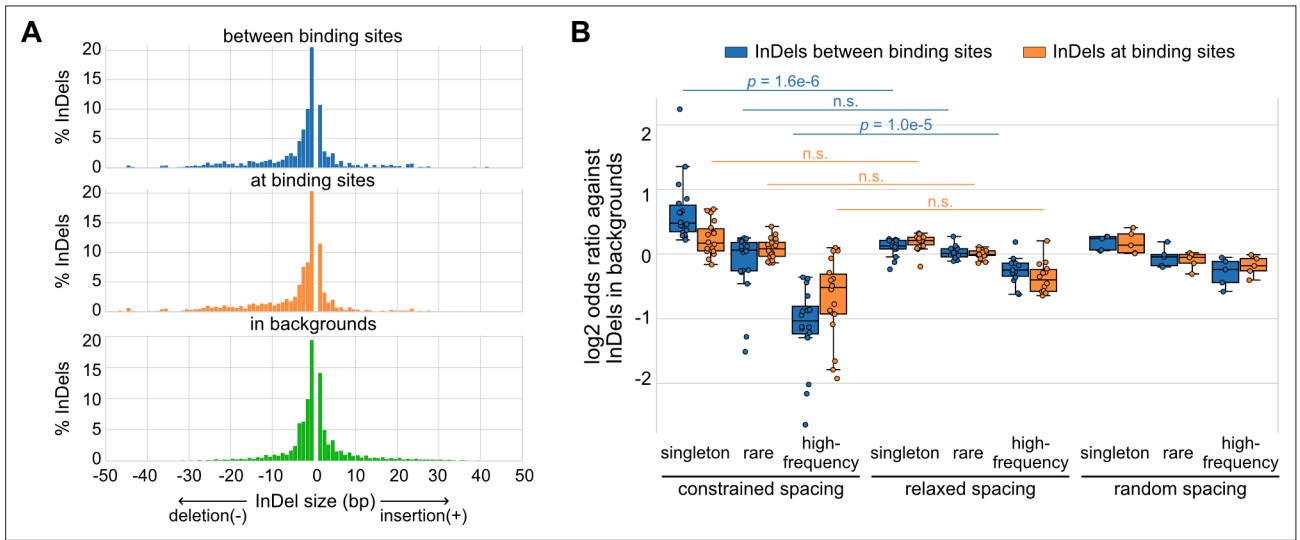

**Figure 2.** Naturally occurring insertions and deletions (InDels) in human populations. (**A**) Size distributions of human InDels within different regions. (**B**) Log2 odds ratios for different categories of InDels. Each dot represents a transcription factor (TF) pair with corresponding spacing relationship. Mann–Whitney U test was used to compare the odds ratios between different spacing relationships. Non-significant (n.s.) if p-value is larger than 0.01.

The online version of this article includes the following source data and figure supplement(s) for figure 2:

**Source data 1.** The numbers and odds ratios of different categories of insertions and deletions (InDels) at or between transcription factor (TF) binding sites.

**Figure supplement 1.** Composition of insertions and deletions (InDels) with different allele frequency (AF) for representative transcription factor (TF) pairs.

**Figure supplement 2.** Log2 odds ratios for insertions and deletions (InDels) separately, complementary to the results of all InDels in *Figure 2B*.

constrained and relaxed spacing groups. On the contrary, singletons were significantly more enriched between the binding sites of TFs with constrained spacing, whereas high-frequency variants were significantly more depleted between these binding sites. We also computed for several TF pairs with random spacing relationships as negative controls and found similar enrichments of InDels like those with relaxed spacing. Since common variants are associated with less deleteriousness and rare variants with more deleteriousness (*Lek et al., 2016*), these findings suggest that InDels between TF binding sites with constrained spacing could be just as damaging as those at binding sites, whereas InDels between TF binding sites with relaxed spacing might have a much weaker effect. This observation is consistent with prior studies that validated significant effects of spacing alterations between TFs with constrained spacing relationships (*Ng et al., 2014*). However, few studies have discussed the effects of InDels on TFs with relaxed spacing, so we specifically focused on relaxed spacing relationships in the rest of the current study.

## Spacing alterations across mouse strains are generally tolerated by relaxed TF binding and promoter and enhancer function

To investigate the regulatory effects of naturally occurring InDels that alter spacing between TFs with relaxed spacing relationships, we leveraged more than 50 million SNPs and 5 million InDels from five genetically diverse mouse strains, including C57BL/6J (C57), BALB/cJ (BALB), NOD/ShiLtJ (NOD), PWK/PhJ (PWK), and SPRET/EiJ (SPRET). The ChIP-seq data of key TFs and histone acetylation and genome-wide transcriptional run-on (GRO-seq) data are available for the bone marrow-derived macrophages (BMDMs) from every mouse strain (*Link et al., 2018a*). We first characterized the spacing relationship between the macrophage LDTFs, PU.1 (encoded by *Spi1*) and C/EBPβ (encoded by *Cebpb*), which have been found to bind in a collaborative manner at the regulatory regions of macrophage-specific genes (*Heinz et al., 2010*). Based on our computational framework for characterizing spacing relationships (*Figure 1A*), these two TFs follow a relaxed spacing relationship independent of their motif orientations (*Figure 3A*; KS p-value < 1e-6). Moreover, both PU.1 and C/EBPβ binding activities quantified by the ChIP-seq tags were indifferent at various spacings (*Figure 3B*).

We then conducted independent comparisons between C57 and one of the other four strains to investigate the effects of spacing alterations caused by natural genetic variation. Most of the natural InDels are less than 5 bp similar to those found in the human population (*Figure 3—figure supplement 1*). We first identified the co-binding sites of PU.1 and C/EBPβ for every strain and then, for each pairwise analysis, pooled the co-binding sites of C57 and a comparison strain to obtain the testing set of regions. Based on the impacts of SNPs and InDels on binding affinity quantified by PWM score or the impacts of InDels on spacing, we categorized the testing regions into the following independent groups: (1) mutated PU.1 motif, (2) mutated C/EBPβ motif, (3) mutated other potentially functional motifs, (4) altered spacing, (5) no motif/spacing effect, and (6) variant free. Potentially functional motifs were identified from PU.1 and C/EBPβ binding sites using MAGGIE (*Shen et al., 2020*), which is a computational tool that finds motifs associated with changes in TF binding (*Figure 3—figure supplement 2*). Considering that PU.1 and C/EBPβ binding could experience changes due to genetic variation mutating other motifs, we grouped these genetic variations to examine their overall effects and simultaneously reach a cleaner group of spacing alterations. The effect of genetic variation was quantified by the log2 fold difference of ChIP-seq tag counts between strains at orthogonal sites (*Figure 3C*). All the four independent comparisons showed that PU.1 binding is most strongly affected by PU.1 motif mutation, followed by C/EBPβ motif mutation and other motif mutation. Spacing alterations have a smaller effect size than any of these motif mutations, but still a relatively larger effect than variants affecting neither binding affinity nor spacing. Despite the moderate effect size of spacing alterations, we found such effect was independent of the size or direction of InDels (*Figure 3D*). On the contrary, changes of PU.1 ChIP-seq tags are strongly correlated with changes of binding affinity measured by changes of PWM scores (*Figure 3D*). In addition, the effects of motif mutation and spacing alteration are not varied by the initial spacing between PU.1 and C/EBPβ motifs (*Figure 3—figure supplement 3*). Similar findings were observed in C/EBPβ binding, except that C/EBPβ motif mutation had the largest effect size and the strongest correlation with C/EBPβ binding activity as expected (*Figure 3E and F*; *Figure 3—figure supplement 3*). Despite that most of the informative genetic variants are located at enhancers and relatively few within promoters, we saw consistent relationships in promoters and enhancers (*Figure 3—figure supplement 4*).

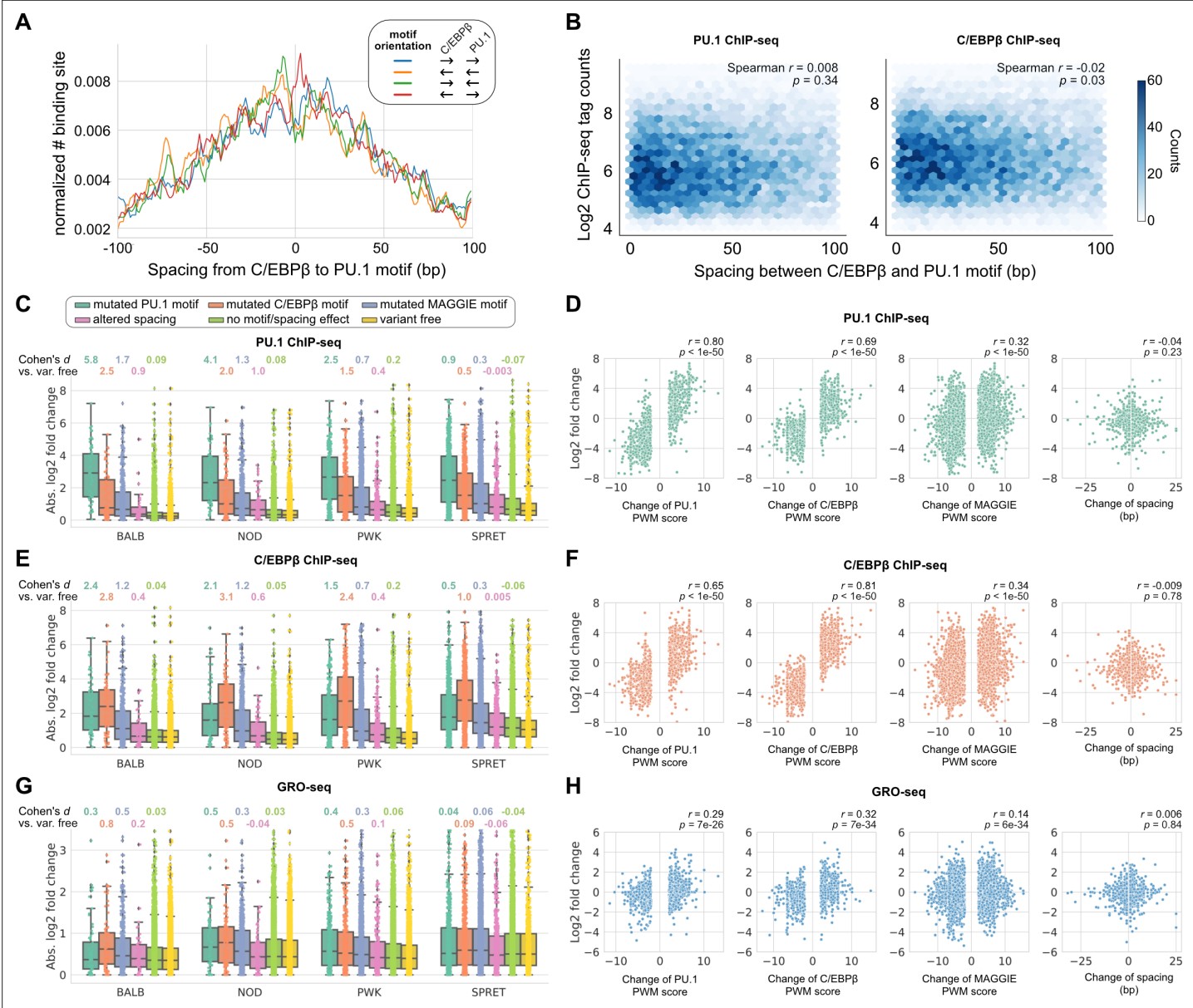

**Figure 3.** Effects of spacing alterations resulting from natural genetic variation across mouse strains. (**A**) Spacing distributions of PU.1 and C/EBPβ binding sites at co-binding sites. (**B**) Density plots showing the relationship between transcription factor (TF) binding activity and motif spacing for the co-binding sites. Log2 chromatin immunoprecipitation sequencing (ChIP-seq) tags were calculated within 300 bp to quantify the binding activity of PU.1 and C/EBPβ. The color gradients represent the number of sites. Spearman's correlation coefficients together with p-values are displayed. (**C, E, G**) Absolute log2 fold changes of ChIP-seq tags between C57 and another strain for (**C**) PU.1 binding, (**E**) C/EBPβ binding, or (**G**) nascent transcripts measured by GRO-seq. Boxplots show the median and quartiles of every distribution. Cohen's d effect sizes comparing against variant-free regions are displayed on top. (**D, F, H**) Correlations between change of spacing or position weight matrix (PWM) score and change of (**D**) PU.1 binding, (**F**) C/EBPβ binding, or (**H**) nascent transcript level. Spearman's correlation coefficients together with p-values are displayed.

The online version of this article includes the following source data and figure supplement(s) for figure 3:

**Source data 1.** Tag fold changes at individual sites for PU.1 chromatin immunoprecipitation sequencing (ChIP-seq).

**Source data 2.** Tag fold changes at individual sites for C/EBPβ chromatin immunoprecipitation sequencing (ChIP-seq).

**Source data 3.** Tag fold changes at individual sites for GRO-seq.

**Source data 4.** Tag fold changes at individual sites for H3K27ac chromatin immunoprecipitation sequencing (ChIP-seq).

**Figure supplement 1.** Size distributions of insertions and deletions (InDels) at PU.1 and C/EBPβ co-binding sites across mouse strains.

**Figure supplement 2.** Functional motifs identified by MAGGIE for different transcription factor (TF) binding.

*Figure 3 continued on next page*

*Figure 3 continued*

**Figure supplement 3.** Absolute log2 fold changes of chromatin immunoprecipitation sequencing (ChIP-seq) tags in relationship with the initial spacing between PU.1 and C/EBPβ motif in the reference mm10 genome.

**Figure supplement 4.** Absolute log2 fold changes of C/EBPβ chromatin immunoprecipitation sequencing (ChIP-seq) tags between C57 and another strain separately showing the distributions of promoters (left) and enhancers (right).

**Figure supplement 5.** Spacing distributions between lineage-determining transcription factors (LDTFs) and signal-dependent transcription factors (SDTFs).

**Figure supplement 6.** Absolute log2 fold changes of chromatin immunoprecipitation sequencing (ChIP-seq) tags between C57 and another strain for lineage-determining transcription factors (LDTFs) and signal-dependent transcription factors (SDTFs).

**Figure supplement 7.** Correlations between changes in transcription factor (TF) binding activity and changes in (**A**) nascent transcription measured by GRO-seq or (**B**) the H3K27ac level measured by chromatin immunoprecipitation sequencing (ChIP-seq).

**Figure supplement 8.** Effects of genetic variation on H3K27ac level.

To investigate whether the effects of altered spacing on PU.1 and C/EBPβ binding can be generalized to hierarchical interactions with signal-dependent TFs (SDTFs), we leveraged the ChIP-seq data of PU.1, the NFκB subunit p65 (encoded by *Rela*), and the AP-1 subunit c-Jun (encoded by *Jun*) for BMDMs treated with the TLR4-specific ligand Kdo2 lipid A (KLA) in the same five strains of mice (*Link et al., 2018a*). Upon macrophage activation with KLA, p65 enters the nucleus and primarily binds to poised enhancer elements that are selected by LDTFs including PU.1 and AP-1 factors (*Heinz et al., 2015*). We observed a relaxed spacing relationship between PU.1 and p65 and between c-Jun and p65 (*Figure 3—figure supplement 5*). In addition, InDels altering motif spacing had a much smaller effect size on TF binding than motif mutations (*Figure 3—figure supplement 6*), consistent with our findings from PU.1 and C/EBPβ.

Although alterations in motif spacing had generally weak effects at the level of DNA binding, it remained possible that changes in motif spacing could influence subsequent steps in enhancer and promoter activation. To examine this, we extended our analysis to nascent transcription measured by GRO-seq (*Core et al., 2008*). Importantly, nascent transcription occurs both at active promoters and enhancers, with enhancer transcription serving as an indicator of enhancer activity (*De Santa et al., 2010*; *Kim et al., 2019*). We leveraged GRO-seq data of untreated BMDMs from the five strains of mice (*Link et al., 2018a*) and calculated the log fold changes of tags at the PU.1 and C/EBPβ co-binding sites for the same pairwise comparisons of strains. Like for TF binding, altered spacing demonstrated weaker effects on nascent transcription than motif mutations (*Figure 3G*), which is consistent with the significant correlations between changes in TF binding and changes in the level of nascent transcripts (*Figure 3—figure supplement 7*). The relative tolerance of spacing alteration was further supported by a weak correlation between changes in GRO-seq tags and the size of InDels, in contrast with a much stronger correlation with changes in binding affinity (*Figure 3H*). Thus, these findings extend the concept of spatial tolerance to the entire ensemble of factors that must be assembled to mediate nascent transcription. Similar relationships were observed for effects of InDels on local acetylation of histone H3 lysine 27 (H3K27ac) (*Figure 3—figure supplement 7*; *Figure 3—figure supplement 8*), which provides an alternative surrogate for enhancer and promoter activity (*Creyghton et al., 2010*).

## Human quantitative trait loci altering spacing between relaxed TFs have small effect sizes

To study the effects of spacing alteration on TF binding and local histone acetylation in human cells, we leveraged the ChIP-seq data of ERG, p65, and H3K27ac in endothelial cells from dozens of individuals (*Stolze et al., 2020*). ERG is an ETS factor that functions as an LDTF in endothelial cells that selects poised enhancers where p65 binds in a hierarchical manner upon interleukin-1β (IL-1β) stimulation (*Hogan et al., 2017*). ERG and p65 follow a relaxed spacing relationship according to our method (*Figure 4A*). Next, we obtained 557 TF binding quantitative trait loci (bQTLs) for ERG, 5,791 bQTLs for p65, 25,621 histone modification QTLs (hQTLs) for H3K27ac in untreated cells, and 21,635 hQTLs for H3K27ac in IL-1β-treated cells (*Stolze et al., 2020*). We further classified bQTLs and hQTLs based on their impacts on binding affinity or spacing: (1) mutated both ERG and p65 (i.e., RELA) motif, (2) mutated ERG motif only, (3) mutated p65 motif only, (4) mutated other potentially functional motifs identified by MAGGIE (*Shen et al., 2020*), (5) altered spacing between ERG and p65 motif, (6) none

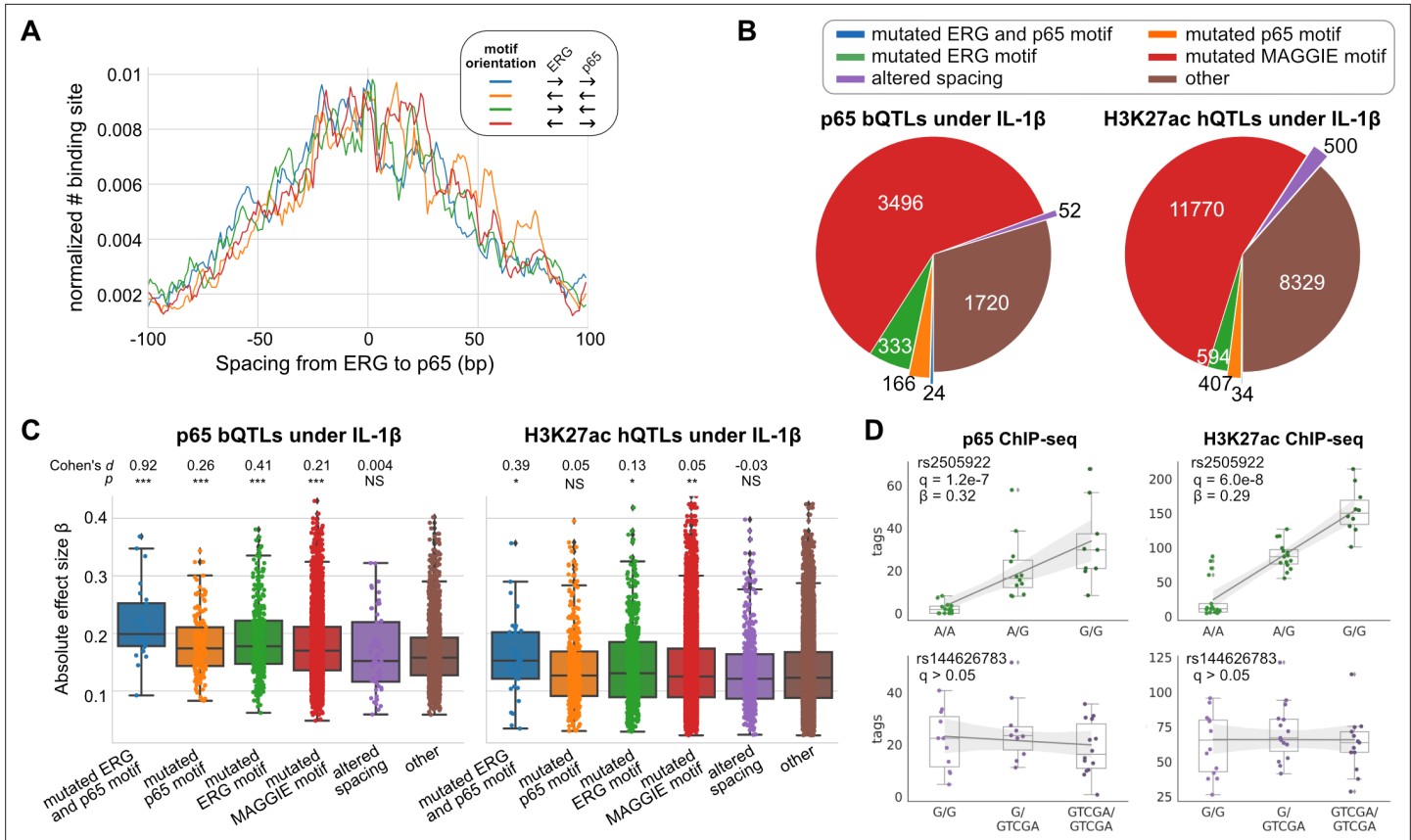

**Figure 4.** Effects of chromatin quantitative trait loci (QTLs) in human endothelial cells. (**A**) Spacing distributions of ERG and p65 binding sites at co-binding sites. (**B**) Classification of chromatin QTLs based on the impacts on motif and spacing. (**C**) Absolute correlation coefficients of different QTLs. Cohen's d and Mann–Whitney U test p-values comparing against the 'other' group are displayed on top. *p < 0.01, **p < 0.001, ***p < 0.0001. (**D**) Example QTLs for large effect size due to ERG motif mutation (upper) and trivial effect due to spacing alteration (lower).

The online version of this article includes the following source data and figure supplement(s) for figure 4:

**Source data 1.** Effect sizes and categorization of p65 binding quantitative trait loci (bQTLs).

**Source data 2.** Effect sizes and categorization of H3K27ac histone modification quantitative trait loci (hQTLs) at IL-1β.

**Source data 3.** Effect sizes and categorization of ERG binding quantitative trait loci (bQTLs).

**Source data 4.** Effect sizes and categorization of H3K27ac histone modification quantitative trait loci (hQTLs) at basal.

**Figure supplement 1.** Functional motifs identified by MAGGIE based on binding quantitative trait loci (bQTLs).

**Figure supplement 2.** Classification of chromatin quantitative trait loci (QTLs) based on the effects on motif and spacing for basal condition.

**Figure supplement 3.** Size distributions of insertions and deletions (InDels) from human endothelial cell donors.

**Figure supplement 4.** Absolute correlation coefficients of different quantitative trait loci (QTLs) for basal condition.

of the above. To find potentially functional motifs, we fed MAGGIE with 100 bp sequences around QTLs before and after swapping alleles at the center (*Figure 4—figure supplement 1*). As a result, only a small portion of bQTLs and hQTLs directly mutates an ERG or p65 motif (*Figure 4B*; *Figure 4—figure supplement 2*). However, such motif mutations are enriched in bQTLs compared to non-QTLs (Fisher's exact p < 1e-4). On the contrary, InDels that alter motif spacing are significantly depleted in p65 bQTLs (Fisher's exact p = 1.3e-15). These InDels from the dozens of individuals are predominantly shorter than 5 bp by following a similar size distribution of those in human populations (*Figure 4—figure supplement 3*). A large proportion of QTLs affect other motifs, implicating the complexity of TF interactions. More than a quarter of the QTLs affect neither binding affinity nor spacing, which can be explained by the high correlation of non-functional variants with functional variants due to linkage disequilibrium.

We further compared the effect sizes of different categories of QTLs. Despite being the minority among QTLs, variants that mutate both ERG and p65 motifs have the strongest effects on both p65 binding and histone acetylation in IL-1β-treated endothelial cells (*Figure 4C*). In comparison, ERG binding and the basal level of histone acetylation are significantly affected by ERG motif mutations in untreated endothelial cells and not by p65 motif mutations, consistent with the hierarchical interaction of p65 only upon IL-1β stimulation (*Figure 4—figure supplement 4*). In both conditions of endothelial cells, spacing alterations have the smaller effect size than motif mutation categories and are not significantly different from likely non-functional variants in the 'other' group. The examples showed a variant being both a p65 bQTL and a H3K27ac hQTL under the IL-1β state due to its impact on an ERG motif, and a 4 bp insertion between ERG and p65 motifs associated with no change in p65 binding or H3K27ac (*Figure 4D*).

## Relaxed TF binding is highly tolerant to synthetic spacing alterations

The generally small effects of InDels occurring between TF pairs exhibiting relaxed spacing relationships raised the question of the robustness and the extent of such tolerance at genomic locations lacking such variation. We addressed this question by using CRISPR/Cas9 editing to introduce

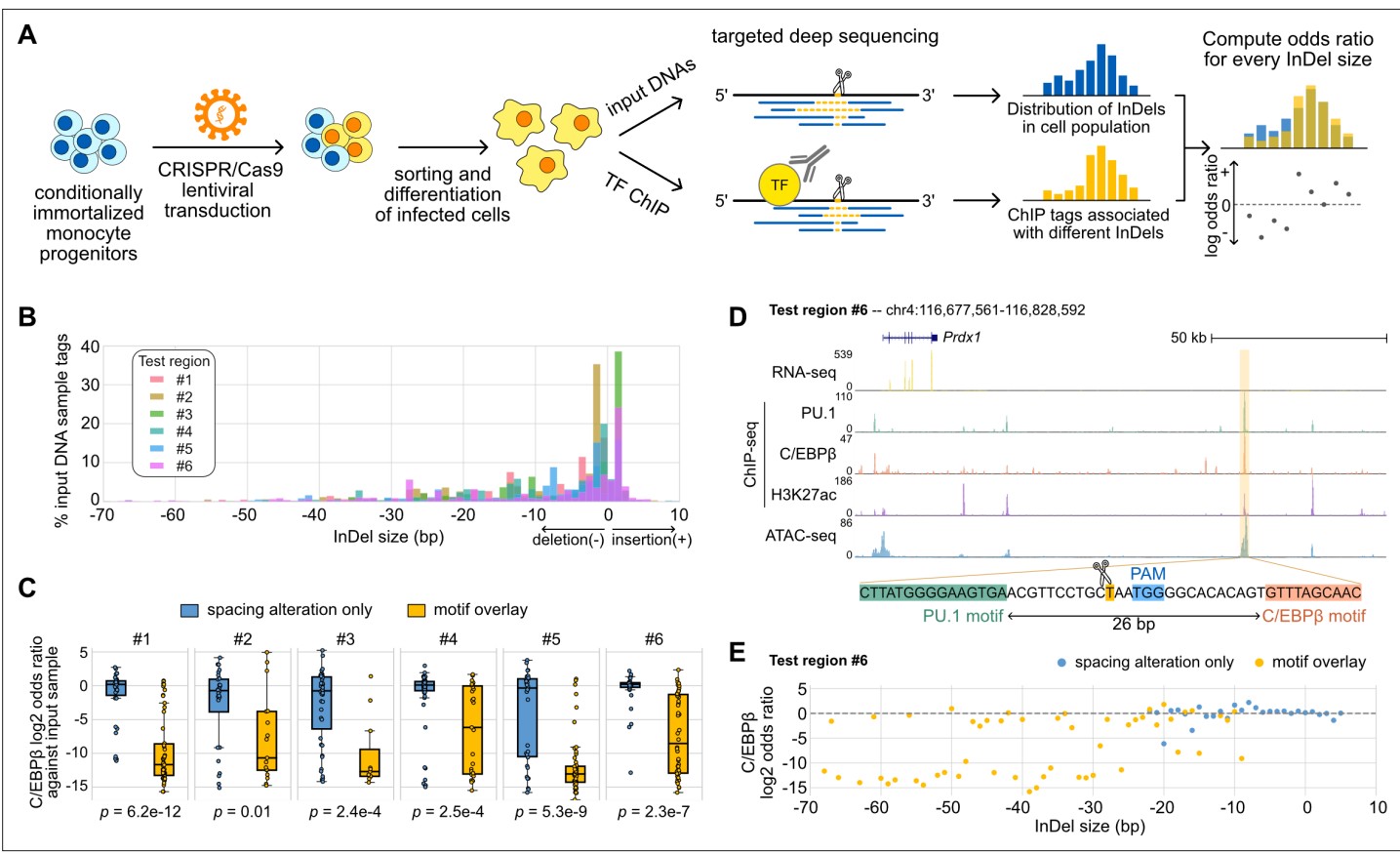

**Figure 5.** Effects of variable sizes of synthetic spacing alterations. (**A**) Schematic for generating and analyzing synthetic spacing alterations. (**B**) The distributions of valid read counts from the input sample based on the InDel sizes of the reads. Negative InDel size indicates deletion, and positive size means insertion. (**C**) Log2 odds ratios by comparing C/EBPβ chromatin immunoprecipitation sequencing (ChIP-seq) reads and input sample reads. Y = 0 indicates where transcription factor (TF) binding has an expected amount of activity. p-Values were based on two-sample t-tests by comparing the InDel groups of each test region. (**D**) Sequencing data of ER-HoxB8 cells at co-binding site of PU.1 and C/EBPβ. Highlighted is test region #6 whose DNA sequence from PU.1 binding site to C/EBPβ binding site is shown. (**E**) Log2 odds ratios of test regions #6 as a function of InDel size.

The online version of this article includes the following source data and figure supplement(s) for figure 5:

**Source data 1.** Raw chromatin immunoprecipitation sequencing (ChIP-seq) tag counts associated with different sizes of insertions and deletions (InDels).

**Figure supplement 1.** Effects of synthetic spacing alterations for test region #1.

**Figure supplement 2.** Effects of synthetic spacing alterations on PU.1 binding.

synthetic InDels between binding sites identified for the LDTFs PU.1 and C/EBPβ in mouse macrophages (*Figure 5A*). We used lentiviral transduction in Cas9-expressing ER-HoxB8 cells, which are conditionally immortalized monocyte progenitors, to introduce gRNAs targeting genomic sequences between the locations of PU.1 and C/EBPβ co-binding. The successfully transduced ER-HoxB8 cells were then sorted and differentiated into macrophages. Since non-homologous DNA repair resulting from the Cas9 nuclease activity would generate a spectrum of InDels in a population of transduced cells, we first measured input DNAs to obtain the distribution of InDels and then compared with TF ChIP-seq tags from deep sequencing, in which the effect of an InDel is reported as the odds ratio of ChIP tags to the input tags. Importantly, the ChIP-seq libraries were prepared by selective amplification of ChIP tags containing the targeted region of interest. Thus, for each region-specific sequence tag that was immunoprecipitated, we could simultaneously determine whether an InDel had been created and its specific length. Each tag is thus cell- and allele-specific.

We tested six PU.1 and C/EBPβ co-binding sites with their original spacing ranging from 26 to 55 bp (*Supplementary file 1*) and quantified the effects of InDels on C/EBPβ binding. Among the six test regions, three of them have supportive evidence from naturally occurring InDels of mouse strains (regions #1, #3, #5) and the other three don't (regions #2, #4, #6). Based on the bioinformatic analysis of the ultra-deep sequencing reads from the input DNA samples, we saw that the CRISPR/Cas9 system generated a wide range of InDels with most deletions being <30 bp and short insertions usually less than 5 bp (*Figure 5B*). It provides longer deletions than natural genetic variations found across mouse strains (*Figure 3—figure supplement 1*) and in human populations (*Figure 2A*). After classifying ChIP-seq reads based on the InDel size and whether the InDel overlaps with any of the PU.1 and C/EBP binding sites, we estimated the effect size of InDels on C/EBPβ binding by calculating the odds ratio between C/EBPβ ChIP-seq reads and input DNA sample reads for every InDel group. We found that InDels altering spacing have significantly weaker effects on C/EBPβ binding in comparison to those overlapping with at least one of the binding sites (*Figure 5C*). For some test regions, the effects of pure spacing alterations are almost negligible, exemplified by test region #6 (*Figure 5D and E*) and test region #1 (*Figure 5—figure supplement 1*). Test region #6 is located near a highly expressed gene *Prdx1* and has strong binding of PU.1 and C/EBPβ binding and strong signals of H3K27ac and chromatin accessibility indicated by ATAC-seq in ER-Hoxb8 cells, which all support its potential regulatory function (*Figure 5D*). The PU.1 and C/EBPβ binding sites at this region are 26 bp apart. In general, spacing alterations ranging from 5 bp increase to 22 bp decrease did not have a strong effect on TF binding, indicated by a log2 odds ratio close to 0 (*Figure 5E*). A small number of outliers were observed at each region where specific InDels resulted in substantial loss of binding (e.g., –20 bp, *Figure 5E*). C/EBPβ binding at these specific InDels was generally discontinuous with 1 bp increments (e.g., –19 and –21 bp, *Figure 5E*). The basis for these highly localized changes in the odds ratio in a small fraction of InDels that alter spacing is unclear. On the contrary, deletions overlapping with the TF binding sites resulted in a general decrease in TF binding activity. Similar results were found at test region #1 where PU.1 and C/EBPβ binding sites are 41 bp apart (*Figure 5—figure supplement 1A*). This *Ly9* enhancer also has a 5 bp insertion between PU.1 and C/EBPβ binding sites in BALB, NOD, and PWK mice, and shows unaffected binding of PU.1 and C/EBPβ in the BMDMs of these strains (*Figure 5—figure supplement 1B*). As a result of the synthetic InDels, the C/EBPβ binding activity was generally unaffected by spacing alterations only, whereas deletions overlapping TF binding sites substantially diminished TF binding (*Figure 5—figure supplement 1C*). We further measured PU.1 binding using ChIP-seq at three out of six test regions and saw general tolerance of synthetic spacing alterations in contrast with significantly weaker PU.1 binding resulted from motif alterations (*Figure 5—figure supplement 2*).

## Discussion

By classifying the genome-wide spacing relationships of 73 co-binding TFs as 'constrained' or 'relaxed', we revealed that relaxed spacing relationships were the dominant pattern of interaction for majority of these factors. Among these factors, approximately half could also participate in constrained spacing relationships with specific TF partners. We confirmed TF pairs known to exhibit constrained relationships (e.g., GATA1-TAL1) and identified previously unreported constrained relationships for additional pairs, including EGR1 and JUND. Overall, this finding of a subset of constrained TF interactions on a genome-wide level is consistent with the locus-specific examples

provided by functional and structural studies of the interferon-β enhanceosome (*Panne, 2008*) and in vivo studies of synthetically modified enhancer elements in *Ciona* (*Farley et al., 2015*). Each of these examples represents genomic regulatory elements in which key TF binding sites are tightly spaced in their native contexts (i.e., 0–9 bp between binding sites). Direct protein-protein interactions are observed between bound TFs at the interferon-β enhanceosome, analogous to interactions defined for cooperative TFs that form ternary complexes (*Morgunova and Taipale, 2017*; *Reményi et al., 2003*). However, unlike the previous in vitro study that identified over 300 TF-TF interactions (*Jolma et al., 2015*), the spacing analyses in our study did not directly consider the possible overlap between TF binding sites. Thus, we are not able to discover constrained TFs that recognize overlapping motifs or distinguish effects of spacing alterations from effects of InDels on overlapping composite motifs.

Our findings based on ChIP-seq data were consistent with the recent in vivo profiling of TF co-occupancy on single DNA molecules, which discovered a lack of association between TF co-occupancy and precise spacing or orientation of motifs (*Sönmezer et al., 2021*). The observation that most TF pairs exhibited relaxed spacing relationships has intriguing implications for the mechanisms by which functional enhancers and promoters are selected from chromatinized DNA. In contrast to ternary complexes of TFs that cooperatively bind to composite elements as a unit, relaxed spacing relationships appear to not require specific protein-protein interactions between TFs for collaborative binding at most genomic locations. Although pioneering TFs necessary for selection of cell-specific enhancers have been reported to recognize their motifs within the context of nucleosomal DNA (*Zaret and Carroll, 2011*), the basis for collaborative binding interactions between TFs with relaxed spacing remains poorly understood.

While the current studies relying on natural genetic variation and mutagenesis experiments concluded clear tolerance of spacing alterations between binding sites of TFs with relaxed spacings, the extent to which this set of binding sites is representative of all regulatory elements is unclear. For example, we observed outliers in which significant differences in TF binding between mouse strains were associated with InDels occurring between TF binding sites. However, the proportion of outliers was generally similar to that observed at genomic regions lacking such InDels, and such strain differences may be driven by distal effects of genetic variation on interacting enhancer or promoter regions (*Hoeksema et al., 2021*; *Link et al., 2018a*). The remarkable tolerance of synthetic InDels at two independent endogenous genomic locations between PU.1 and C/EBPβ binding sites strongly support the generality of relaxed binding interactions for these two proteins. Intriguingly, while the densities of C/EBP binding sites increase with decreasing distance to PU.1 binding sites over a 100 bp range (*Figure 3A*), deletions from 1 to >30 bp between PU.1-C/EBPβ pairs did not result in improved binding. Instead, relatively constant binding was observed with progressive deletions bringing two binding sites close together until the deletions started to cause mutations in one or both motifs. The lack of requirement for exact spacing and remarkable tolerance of spacing alterations by TFs with relaxed spacing could potentially associate with the high turnover of TF binding sites found by previous studies (*Vierstra et al., 2014*), although further investigation would be needed to establish this association. A limitation of our studies is that few and relatively short insertions were obtained, preventing conclusions as to the extent to which increases in spacing are tolerated.

In concert, the present studies provide a basis for estimation of the potential phenotypic consequences of naturally occurring InDels in non-coding regions of the genome. The majority of naturally occurring InDels are less than 5 bp in length. In nearly all cases, InDels of this size range between binding sites for TFs that have relaxed binding relationships are unlikely to alter TF binding and function, and InDels of much greater length are frequently tolerated. In contrast, InDels between binding sites for TFs that have constrained binding relationships have the potential to result in biological consequences. Application of these findings to the interpretation of non-coding InDels that are associated with disease risk will require knowledge of the relevant cell type in which the InDel exerts its phenotypic effect and the types of TF interactions driving the selection and function of the affected regulatory elements.

## Materials and methods

**Key resources table**

| Reagent type (species) or resource | Designation | Source or reference | Identifiers | Additional information |
|---|---|---|---|---|
| Strain, strain background (*Mus musculus*, male) | B6(C)-Gt(ROSA)26Sor[em1.1(CAG-cas9*,-EGFP)Rsky]/J | Jackson Laboratory | Stock No: 028555 RRID:IMSR_JAX:028555 | |
| Cell line (*Mus musculus*) | Cas9-expressing ER-HoxB8 cells | This paper | | Gifted from Dr David Sykes |
| Cell line (human) | Lenti-X 293T cells | Clontech | Cat#: 632180 RRID:CVCL_4401 | |
| Transfected construct (retrovirus) | Murine stem cell virus-based vector for ER-HoxB8 | Massachusetts General Hospital, Boston, MA | | Gifted from Dr David Sykes |
| Transfected construct (retrovirus) | lentiGuide-puro | Addgene | Cat#: 52963 | |
| Transfected construct (retrovirus) | psPAX2 | Addgene | Cat#: 12260 | |
| Transfected construct (retrovirus) | pVSVG | Addgene | Cat#: 138479 | |
| Antibody | PU.1/Spi1 (rabbit polyclonal) | Santa Cruz | Cat#: sc-352X RRID:AB_632289 | (1 µL) |
| Antibody | C/EBPβ (rabbit polyclonal) | Santa Cruz | Cat#: sc-150 RRID:AB_2260363 | (10 µL) |
| Antibody | H3K27ac (rabbit polyclonal) | Active Motif | Cat#: 39135 RRID:AB_2614979 | (2 µL) |
| Recombinant DNA reagent | NEBNext 2× High Fidelity PCR Master Mix | NEB | Cat#: M0541 | |
| Sequence-based reagent | Locus-specific Nextera hybrid primer | This paper | PCR primers | Sequences included in **Supplementary file 1** |
| Sequence-based reagent | Nextera index primer | This paper | PCR primers | Sequences included in **Supplementary file 1** |
| Peptide, recombinant protein | Recombinant Mouse IL-3 | Peprotech | Cat#: 213–13 | |
| Peptide, recombinant protein | Recombinant Mouse IL-6 | Peprotech | Cat#: 216–16 | |
| Peptide, recombinant protein | Recombinant Mouse SCF | Peprotech | Cat#: 250–03 | |
| Peptide, recombinant protein | Recombinant Mouse GM-CSF | Peprotech | Cat#: 315–03 | |
| Peptide, recombinant protein | Mouse M-CSF | Shenandoah Biotech | Cat#: 200–08 | |
| Commercial assay or kit | Direct-zol RNA MicroPrep kit | Zymo Research | Cat#: R2062 | |
| Commercial assay or kit | Qubit dsDNA HS Assay Kit | Thermo Fisher Scientific | Cat#: Q32851 | |
| Commercial assay or kit | Nextera DNA Library Preparation Kit | Illumina | Cat#: 15028212 | |
| Commercial assay or kit | ChIP DNA Clean & Concentrator | Zymo Research | Cat#: D5205 | |
| Commercial assay or kit | NEBNext Ultra II Library Preparation Kit | NEB | Cat#: E7645L | |
| Chemical compound, drug | LentiBlast Transduction Reagent | OZ Biosciences | Cat#: LB00500 | |
| Chemical compound, drug | Ficoll-Paque-Plus | Sigma-Aldrich | Cat#: GE17-1440-02 | |
| Chemical compound, drug | RPMI-1640 | Corning | Cat#: 10–014-CV | |

| Reagent type (species) or resource | Designation | Source or reference | Identifiers | Additional information |
|---|---|---|---|---|
| Chemical compound, drug | DMEM high glucose | Corning | Cat#: 10–013-CV | |
| Chemical compound, drug | FBS | Omega Biosciences | Cat#: FB-12 | |
| Chemical compound, drug | 100× Penicillin/ Streptomycin + L-glutamine | Gibco | Cat#: 10378–016 | |
| Chemical compound, drug | β-Estradiol | Sigma-Aldrich | Cat#: E2758 | |
| Chemical compound, drug | G418 | Thermo Fisher | Cat#: 10131035 | |
| Chemical compound, drug | Polybrene | Sigma-Aldrich | Cat#: H9268 | |
| Chemical compound, drug | Fibronectin | Sigma-Aldrich | Cat#: F0895 | |
| Chemical compound, drug | Poly-D-lysin | Sigma-Aldrich | Cat#: DLW354210 | |
| Chemical compound, drug | X-tremeGENE HP DNA Transfection Reagent | Sigma-Aldrich | Cat#: 6366546001 | |
| Chemical compound, drug | Formaldehyde | Thermo Fisher Scientific | Cat#: BP531-500 | |
| Chemical compound, drug | Dynabeads Protein A | Invitrogen | Cat#: 10002D | |
| Chemical compound, drug | SpeedBeads magnetic carboxylate modified particles | Sigma-Aldrich | Cat#: GE65152 105050250 | |
| Chemical compound, drug | Dynabeads MyOne Streptavidin T1 | Invitrogen | Cat#: 65602 | |
| Software, algorithm | CHOPCHOP | CHOPCHOP (https://chopchop.cbu.uib.no/) | RRID:SCR_015723 | |
| Software, algorithm | Bowtie2 | Bowtie2 (http://bowtie-bio.sourceforge.net/bowtie2/index.shtml) | RRID:SCR_016368 | Version 2.3.5.1 |
| Software, algorithm | STAR | STAR (https://github.com/alexdobin/STAR) | RRID:SCR_004463 | Version 2.5.3 |
| Software, algorithm | HOMER | HOMER (https://homer.ucsd.edu/homer/) | RRID:SCR_010881 | Version 4.9.1 |
| Software, algorithm | MAGGIE | MAGGIE (https://github.com/zeyang-shen/maggie) | RRID:SCR_021903 | Version 1.1 |
| Software, algorithm | IDR | IDR (https://www.encodeproject.org/software/idr/) | RRID:SCR_017237 | Version 2.0.3 |
| Software, algorithm | MMARGE | MMARGE (https://github.com/vlink/marge) | RRID:SCR_021902 | Version 1.0 |

## Sequencing data processing

We downloaded two replicates for each TF ChIP-seq data from ENCODE data portal (*Davis et al., 2018*). The mouse BMDM data and the human endothelial cell data were downloaded from the GEO database with accession number GSE109965 (*Link et al., 2018a*) and GSE139377 (*Stolze et al., 2020*), respectively. We mapped ChIP-seq and ATAC-seq reads using Bowtie2 v2.3.5.1 with default parameters (*Langmead and Salzberg, 2012*) and mapped RNA-seq reads using STAR v2.5.3 (*Dobin et al., 2013*). All the human data downloaded from ENCODE were mapped to the hg38 genome. Data from C57BL/6J mice were mapped to the mm10 genome. Data from other mouse strains and endothelial cell data from different individuals were mapped to their respective genomes built by MMARGE v1.0 (*Link et al., 2018b*). More details are described below.

Based on the mapped ChIP-seq data, we called peaks using HOMER v4.9.1 (*Heinz et al., 2010*). For data with replicates including ENCODE data and mouse data, we first called unfiltered 200 bp peaks using HOMER 'findPeaks' function using parameters '-style factor -L 0 C 0 -fdr 0.9 -size 200' and then ran IDR v2.0.3 with default parameters (*Li et al., 2011*) to obtain reproducible peaks. For data without replicates including human endothelial cell data and ER-HoxB8 ChIP-seq data, we called peaks using HOMER 'findPeaks' with the default setting and parameters '-style factor -size 200'.

Activity of TF binding and nascent transcription was quantified by the ChIP-seq and GRO-seq tag counts, respectively, within 300 bp around peak centers and normalized by library size using HOMER 'annotatePeaks.pl' script with parameters '-norm 1e7 -size –150,150'. Activity of promoter and enhancer was quantified by normalized H3K27ac ChIP-seq tags within 1000 bp regions around TF peak centers using parameters '-norm 1e7 -size –500,500'.

## TF binding site identification

Given a DNA sequence with the same length of a TF binding motif, we calculated a PWM score, or sometimes called motif score, by the dot product between the motif PWM and the sequence vector using Biopython package (*Cock et al., 2009*). The PWMs of all the TFs in this study were obtained from either the JASPAR database (*Fornes et al., 2020*) or de novo motif analysis using HOMER 'findMotifsGenome.pl -size 200 -mask' with random backgrounds (*Heinz et al., 2010*) if unavailable in the JASPAR database. For de novo motifs, the ultimate motif of each TF used for identifying TF binding sites was manually selected from the top three significant motifs that look like motifs of other TFs within the same TF family available in JASPAR. The final motifs are listed in *Supplementary file 1*. TFs without a confident motif based on the criteria above or ending up with less than 2000 ChIP-seq peaks with at least one confident TF binding site based on the pipeline below have been excluded from the current study.

The original PWMs were first trimmed to keep only the core motifs starting from the first position of information content greater than 0.3 to the last position of information content greater than 0.3 (*Ng et al., 2014*). A valid TF binding site was identified by having a PWM score passing a false positive rate (FPR) of 0.1% based on a score distribution generated from 25% composition of A, C, G, T using Biopython (*Cock et al., 2009*) and a location within 50 bp close to the ChIP-seq peak center. Spacing is computed edge-to-edge between two TF binding sites at co-binding sites. If there are multiple valid motifs for one or both TFs, we computed the spacing between all possible combinations of valid motifs. For TFs that share a core motif, we applied the same pipeline to each TF separately, which may or may not identify the same binding sites, but we only included non-overlapping binding sites in our downstream calculation of spacing.

## Characterization of spacing relationships

To test for the constrained spacing relationship between a given pair of TFs, we first generated their spacing distribution at single-base-pair resolution ranging from –100 to +100 bp and then used Monte Carlo procedures to identify significant 'spikes' in the spacing distribution based on point-to-point slopes. The slope of position $i$ is computed using the following formula:

$$S_i = \frac{2N_i - N_{i-1} - N_{i+1}}{2}, \ i \in [-99, 99]$$

$S_i$ is the average of single-step forward and backward slope at spacing equal to $i$ bp, and $N_i$ represents the value of position $i$ on the spacing distribution (i.e., the number of TF binding sites with

spacing *i* bp). We then used Monte Carlo procedures to obtain an empirical p-value to represent the statistical significance of a slope to be considered as a 'spike'. Specifically, we randomly sampled 1000 integers between 0 and 100, calculated a distribution, and obtained all the slopes using the formula above. We repeated this process by 1000 times and summarized all the slopes in an aggregated distribution. p-Value is determined based on the percentile rank of testing value on the aggregated distribution. p-Value smaller than 6.25e-05 (familywise error rate = 0.05/200/4) is called significant, indicating a 'spike' is found.

To test for the relaxed spacing relationship, we used KS test to compare a spacing distribution to the random distribution. We randomly sampled integers between –100 and 100 to match the same size of the testing spacing distribution and then tested the spacing distribution against the distribution of the random integers to obtain a p-value. We repeated the above process 100 times and reported the average p-value. To compare spacing relationships in repetitive versus nonrepetitive regions, we first used repeat annotations from HOMER to divide our ChIP-seq peaks into repetitive and nonrepetitive regions and then repeated the above procedures.

## Categorization of gnomAD variants based on AF

We obtained InDels from gnomAD v3.1 (*Karczewski et al., 2020*). These gnomAD variants were mapped to TF co-binding sites, specifically with two TF binding sites and their intermediate sequences. For TF pairs with constrained spacing relationships, we only kept the co-binding sites with the identified constrained spacing ±2 bp. To account for region-by-region variation in selective pressure, we also overlapped variants with 100 bp upstream and 100 bp downstream background regions outside of TF co-binding sites. For each co-binding site, we categorized InDels into high-frequency variants (AF>0.01%), rare variants (AF <0.01%, AC >1), and singletons (AC = 1) and computed the odds ratios to find associations between a certain category of InDels (e.g., singletons) and certain regions (e.g., between motifs with constrained spacing).

## Genetic variation processing and genome building

Genetic variation of the five mouse strains was obtained from *Keane et al., 2011*, and that of the human individuals from which endothelial cell data were generated was derived from *Stolze et al., 2020*. We used MMARGE v1.0 with default variant filters (*Link et al., 2018b*) to build separate genomes for each mouse strain and human individual. The sequencing data from different samples were respectively mapped to the corresponding genomes and were then shifted to a common reference genome using MMARGE 'shift' function to facilitate comparison at homologous regions. The reference genome is mm10 for mouse strains and hg19 for human individuals.

## Motif mutation analysis

We used MAGGIE v1.0 (*Shen et al., 2020*) to identify functional motifs for TF binding. To prepare the inputs into MAGGIE based on the mouse strains data, we adapted a similar strategy as described in *Shen et al., 2020*. In brief, we conducted pairwise orthogonal comparisons of TF peaks between each possible pair of the five strains to find strain-differential peaks. We then extracted pairs of 200 bp sequences around the centers of the differential peaks from the genomes of two comparative strains, the ones with TF binding as positive sequences paired with those without TF binding as negative sequences. For the QTLs of human endothelial cells, MAGGIE can directly work with a VCF file of QTLs with effect size and effect direction indicated in a column of the file. We ran MAGGIE separately for each type of QTLs and reported the significant motifs together with their p-values, which passed false discovery rate (FDR) < 0.05 after the Benjamini–Hochberg controlling procedure.

## Categorization of genetic variation based on impacts on motif or spacing

We categorized genetic variation based on its impact on binding affinity or spacing. Motif mutations caused by SNPs or InDels were defined by at least two-bit difference in the PWM score, which is equivalent to approximately 4-fold difference in the binding likelihood. To classify genetic variation that mutates other functional motifs identified by MAGGIE, we required at least one of the MAGGIE motifs differed by at least two bits in the PWM score. InDels were first classified into motif mutation categories if eligible before being considered for spacing alteration. Therefore, spacing alterations

were InDels between target motifs without any motif mutations. Variants fitting neither motif mutation nor spacing alteration were gathered in a separate group as a control. Another control category during analysis of mouse strains data was defined by ChIP-seq peaks that have no genetic variation between strains.

## Statistical testing of effect size

To estimate the effects of genetic variation on different readouts, we computed the log2 fold changes of corresponding sequencing tags for every pairwise comparison between C57 and one of the other strains. The effect sizes of QTLs were directly pulled from the source literature. We conducted Mann–Whitney U tests between the control variant category and another category to find significant differences in their distributions. Cohen's d (*Sullivan and Feinn, 2012*) was further calculated between two distributions. For an easier comparison of general effect size, we took the absolute values before calculating Cohen's d. Spearman's correlation coefficient together with statistical significance was calculated to find correlations between log2 fold changes of tags and TF features.

## ER-HoxB8 cell-derived macrophage culture and CRISPR knockout

Bone marrow cells were isolated from femurs and tibias of a Cas9-expressing transgenic mouse (Jackson Laboratory, No. 028555). Murine stem cell virus-based expression vector for ER-HoxB8 was gifted from Dr David Sykes (Massachusetts General Hospital, Boston, MA). Cas9-expressing ER-HoxB8 conditionally immortalized myoid progenitor cells were generated following established protocols (*Wang et al., 2006*). In brief, bone marrow cells were purified with a Ficoll gradient (Ficoll-Paque-Plus, Sigma-Aldrich) and resuspended in RPMI 1640 containing 10% FBS, 1% penicillin/streptomycin, and 10 ng/ml each of murine SCF, IL-3, and IL-6 (PeproTech). After 48 hr culture, $2.5 \times 10^5$ cells in 1 ml were transduced with 2 ml of ER-HoxB8 retrovirus (in DMEM with 30% FBS) containing 0.5 μl/ml LentiBlast A (OZ Biosciences), 2.5 μl/ml LentiBlast B (OZ Biosciences) and 8 μg/ml polybrene (Sigma-Aldrich) in a well of fibronectin (Sigma-Aldrich)-coated six-well culture plates and centrifuged at 1000 g for 90 min at 22°C. After transduction, 6 ml of ER-HoxB8 cell culture media (RPMI 1640 supplemented with 10% FBS, 1% penicillin/streptomycin, 0.5 μM β-estradiol (Sigma-Aldrich), and 20 ng/ml murine GM-CSF (PeproTech)) were added and an additional half-media exchange with ER-Hoxb8 media performed the next day. Transduced cells were selected with G418 (Thermo Fisher) at 1 mg/ml for 48 hr. Thereafter, cells were maintained in ER-HoxB8 cell culture media and confirmed its identity by RNA-seq. For baseline ATAC-seq and ChIP-seq of ER-HoxB8 cells prior to gRNA transduction, cells were washed twice with PBS, plated at a density of $3 \times 10^6$ cells per 10 cm culture plate, and differentiated into macrophages in DMEM with 10% FBS, 1% pencillin/streptomycin, and 17 ng/ml M-CSF (Shenandoah) for 7 days with two culture media exchanges. Differentiated cells were washed twice with PBS and collected for sequencing experiments.

Guide RNA lentiviruses were prepared as previously described (*Fonseca et al., 2019*) with modifications as follows. LentiGuide-mCherry was generated by modifying lentiGuide-puro (Addgene) to remove a puromycin-resistant gene and replace it with mCherry. gRNA sequences directed between the PU.1 and C/EBP binding sites (and one each directed toward the binding site itself) were designed with CHOPCHOP web tool for genome engineering (*Labun et al., 2019*). One CRISPR gRNA oligonucleotide was inserted for each target via PCR into a BsmBI cleavage site. A list of gRNA targets used in this article is shown in *Supplementary file 1*. Lenti-X 293T cells (Clontech) were seeded in poly-D-lysin (Sigma-Aldrich) coated 10 cm tissue culture plates at a density of 3.5 million cells per plate in 10 ml of DMEM containing 10% FBS and 1% penicillin/streptomycin, and then incubated overnight at 37°C. After replacement of the media to 6 ml of DMEM containing 30% FBS, plasmid DNAs (5 μg of LentiGuide-mCherry, 3.75 μg of psPAX2, and 1.25 μg of pVSVG) were transfected into LentiX-293T cells using 20 μl of X-tremeGENE HP DNA Transfection Reagent (Roche) at 37°C overnight. The media was replaced with DMEM containing 30% FBS and 1% penicillin/streptomycin, and then cultured at 37°C overnight. The supernatant was filtrated with 0.45 μm syringe filters and used as lentivirus media. Cell culture media was replaced, and virus was collected again after 24 hr. $1 \times 10^6$ Cas9-expressing ER-HoxB8 cells were transduced with virus in 2 ml of lentivirus media and 1 ml of ER-HoxB8 cell media containing 0.5 μl/ml LentiBlast A, 2.5 μl/ml LentiBlast B, and 8 μg/ml polybrene in a well of fibronectin-coated six-well culture plates and centrifuged at 1000 g for 90 min at 22°C. After the transduction, 6 ml of ER-HoxB8 cell media was added to each well. Half of the media was exchanged the next day

and in the following days, cells were expanded and passaged. After 5 days, 250,000 successfully transduced cells (indicated by mCherry fluorescence) for each gRNA were sorted by FACS using a Sony MA900. After FACS, cells were expanded in ER-HoxB8 culture media. Differentiation into macrophages was carried out as above in DMEM supplemented with M-CSF.

## RNA-seq library preparation

Total RNA was isolated from cells and purified using Direct-zol RNA Microprep columns according to the manufacturer's instructions (Zymo Research); 500 ng of total RNA were used to prepare sequencing libraries from polyA enriched mRNA as previously described (*Link et al., 2018a*). Libraries were PCR-amplified for 14 cycles, size selected using SpeedBeads (Sigma-Aldrich), quantified by Qubit dsDNA HS Assay Kit (Thermo Fisher Scientific), and 75 bp single-end sequenced on a HiSeq 4000 (Illumina).

## ATAC-seq library preparation

ATAC-seq libraries were prepared as previously described (*Hoeksema et al., 2021*). In brief, $5 \times 10^5$ cells were lysed at room temperature in 50 µl ATAC lysis buffer (10 mM Tris-HCl, pH 7.4, 10 mM NaCl, 3 mM $MgCl_2$, 0.1% IGEPAL CA-630) and 2.5 µl DNA Tagmentation Enzyme mix (Nextera DNA Library Preparation Kit, Illumina) was added. The mixture was incubated at 37°C for 30 min and subsequently purified using the ChIP DNA Clean & Concentrator kit (Zymo Research) as described by the manufacturer. DNA was amplified using the Nextera Primer Ad1 and a unique Ad2.n barcoding primers using NEBNext High-Fidelity 2× PCR MM for 8–14 cycles. PCRs were size selected using TBE gels for 175–350 bp and DNA eluted using gel diffusion buffer (500 mM ammonium acetate, pH 8.0, 0.1% SDS, 1 mM EDTA, 10 mM magnesium acetate) and purified using ChIP DNA Clean & Concentrator (Zymo Research). Samples were quantified by Qubit dsDNA HS Assay Kit (Thermo Fisher Scientific) and 75 bp single-end sequenced on HiSeq 4000 (Illumina).

## Crosslinking for ChIP-seq

For PU.1, C/EBPβ, and H3K27ac ChIP-seq, culture media was removed, and plates were washed once with PBS and then fixed for 10 min with 1% formaldehyde (Thermo Fisher Scientific) in PBS at room temperature. Reaction was then quenched by adding glycine (Thermo Fisher Scientific) to 0.125 M. After fixation, cells were washed once with cold PBS and then scraped into supernatant using a rubber policeman, pelleted for 5 min at 400× *g* at 4°C. Cells were transferred to Eppendorf DNA LoBind tubes and pelleted at 700× *g* for 5 min at 4°C, snap-frozen in liquid nitrogen, and stored at –80°C until ready for ChIP-seq protocol preparation.

## Chromatin immunoprecipitation

ChIP was performed in biological replicates as described previously (*Hoeksema et al., 2021*). Samples were sonicated using a probe sonicator in 500 µl lysis buffer (10 mM Tris/HCl pH 7.5, 100 mM NaCl, 1 mM EDTA, 0.5 mM EGTA, 0.1% deoxycholate, 0.5% sarkozyl, 1× protease inhibitor cocktail). After sonication, 10% Triton X-100 was added to 1% final concentration and lysates were spun at full speed for 10 min; 1% was taken as input DNA, and immunoprecipitation was carried out overnight with 20 µl Protein A Dynabeads (Invitrogen) and 2 µg specific antibodies for C/EBPβ (Santa Cruz, sc-150), PU.1 (Santa Cruz, sc-352X), and H3K27ac (Active Motif, 39135). Beads were washed three times each with wash buffer I (20 mM Tris/HCl, 150 mM NaCl, 0.1% SDS, 1% Triton X-100, 2 mM EDTA), wash buffer II (10 mM Tris/HCl, 250 mM LiCl, 1% IGEPAL CA-630, 0.7% Na-deoxycholate, 1 mM EDTA), TE 0.2% Triton X-100 and TE 50 mM NaCl and subsequently resuspended 25 µl 10 mM Tris/HCl pH 8.0% and 0.05% Tween-20. ChIP-seq libraries were prepared on the Dynabeads as described below. For locus-specific enrichment ChIP-seq, bead complex was resuspended in 50 µl 1% SDS-TE. Four µl ProtK, 4 µl RNase A, 3 µl 5 M NaCl were added to these and the input samples and incubated at 50°C for 1 hr, reverse crosslinked at 65°C overnight and then eluted from the beads.

## ChIP-seq library preparation

ChIP libraries were prepared while bound to Dynabeads using NEBNext Ultra II Library preparation kit (NEB) using half reactions. DNA was polished, polyA-tailed, and ligated after which dual UDI (IDT) or single (Bioo Scientific) barcodes were ligated to it. Libraries were eluted and crosslinks reversed by adding to the 46.5 µl NEB reaction 16 µl water, 4 µl 10% SDS, 4.5 µl 5 M NaCl, 3 µl 0.5 M EDTA, 4 µl

0.2 M EGTA, 1 µl RNAse (10 mg/ml), and 1 µl 20 mg/ml proteinase K, followed by incubation at 55°C for 1 hr and 75°C for 30 min in a thermal cycler. Dynabeads were removed from the library using a magnet and libraries were cleaned up by adding 2 µl SpeedBeads (Sigma-Aldrich) in 124 µl 20% PEG 8000/1.5 M NaCl, mixing well, then incubating at room temperature for 10 min. SpeedBeads were collected on a magnet and washed two times with 150 µl 80% ethanol for 30 s. Beads were collected and ethanol removed following each wash. After the second ethanol wash, beads were air dried and DNA eluted in 12.25 µl 10 mM Tris/HCl pH 8.0% and 0.05% Tween-20. DNA was amplified by PCR for 14 cycles in a 25 µl reaction volume using NEBNext Ultra II PCR master mix and 0.5 µM each Solexa 1GA and Solexa 1 GB primers. Libraries were size selected using TBE gels for 200–500 bp and DNA eluted using gel diffusion buffer (500 mM ammonium acetate, pH 8.0, 0.1% SDS, 1 mM EDTA, 10 mM magnesium acetate) and purified using ChIP DNA Clean & Concentrator (Zymo Research). Sample concentrations were quantified by Qubit dsDNA HS Assay Kit (Thermo Fisher Scientific) and 75 bp single-end sequenced on HiSeq 4000.

### Biotin-mediated locus-specific enrichment ChIP-seq library preparation

After performing the target-specific ChIPs, we performed an initial PCR for locus-specific amplicon enrichment using NEBNext 2× High Fidelity PCR MM (NEB) and 5'-biotinylated stub adapter primers specific to appropriate genomic regions to be interrogated (*Supplementary file 1*). Initial hotstart/denaturation at 98°C for 30 s was followed by 10 cycles of amplification (98°C for 15 s, 65–67°C for 15 s, 72°C for 30 s) and then a final elongation at 72°C for 5 min. After this, we performed a 0.7× AmpureXP clean-up and eluted in 20 µl 0.5× TT (5 mM Tris pH 8.0 + 0.025% Tween-20). Dynabeads MyOne Streptavidin T1 beads were then washed in 1× Wash Binding Buffer (WBB, 2× WBB: 10 mM Tris-HCl [pH 7.5], 1 mM EDTA, 2 M NaCl, 0.1% Tween) and resuspended beads at 20 µl per sample in 2× WBB. Twenty µl prepared Dynabeads MyOne Streptavidin T1 beads (in 2× WBB) were then added to clean up 20 µl 0.5× TT PCR fragments, mixed and incubated for 60 min at room temperature with mild shaking. After this, beads were collected on a magnet and washed twice with 150 µl 1× WBB and once with 180 µl TET (TE +0.05% Tween-20). Finally, beads were resuspended in 25 µl 0.5× TT and on bead PCR for addition of Illumina-specific adapters and 10 bp Unique Dual Indexes (UDIs) using NEBNext 2× High Fidelity PCR MM (NEB) and 25 PCR cycles was performed (*Supplementary file 1*). Libraries were size selected using TBE gels for 300–500 bp and DNA eluted using gel diffusion buffer and purified using ChIP DNA Clean & Concentrator (Zymo Research). Samples were quantified by Qubit dsDNA HS Assay Kit (Thermo Fisher Scientific) and 150 bp paired-end sequenced on NextSeq 500 (Illumina).

### Analysis of variable InDels from CRISPR experiments

We mapped the reads to the target regions using the local alignment mode of Bowtie2 v2.3.5.1 (*Langmead and Salzberg, 2012*). To allow for InDels with tens of bases, we reduced the gap extend penalty and increased the gap open penalty so that the gaps could be long but not occur at multiple locations. Here are the adjusted parameters used in our mapping process: `--local --rdg 10,1 --rfg 10,1`. The mapped reads with gaps or InDels at unexpected locations rather than the Cas9 cut sites were removed. This step filtered out approximately 1% of the total reads (*Supplementary file 1*). The remaining reads were grouped based on the InDel size and whether the InDel overlaps with any of the PU.1 and C/EBPβ binding sites. Tag counts were used to quantify binding activity. InDel groups taking up less than 0.05% of the input sample reads were filtered out. TF binding associated with each InDel group was computed by the odds ratio between TF ChIP-seq tags and input DNA sample tags: (TF tags for an InDel group/rest of TF tags)/(input tags for the same InDel group/rest of input tags).

### Cell lines

These studies made use of two cell lines. The major cell line used was Cas9-expressing ER-HoxB8 cells. This cell line was generated in our laboratory from primary bone marrow progenitor cells and its identity and phenotype are confirmed by RNA-seq. Given the origin of these cells and continuous validation by RNA-seq, we do not routinely test them for mycoplasma. In addition, because of the nature of the experimental design for the use of these cells, in which we are analyzing ChIP tags within a single population of cells, the presence of mycoplasma infection would not alter the outcome unless

it changed the expression of the TFs being ChIP'd. This is clearly not the case based on our RNA-seq and ChIP-seq results.

The second cell line corresponded to 293T cells. These cells were used to generate lentivirus for transduction of the ER-HoxB8 cells. We acknowledge that these cells have the potential to be infected with mycoplasma and for viral supernatants to transmit this to the ER-HoxB8 cells. However, even if this were to be the case, it would not alter the conclusions of the experiment using the ER-HoxB8 cells for the reasons noted above.

## Acknowledgements

The authors would like to thank J Collier for technical assistance, the IGM core for library sequencing, L Van Ael for assistance with manuscript preparation. These studies were supported by NIH grants DK091183 and HL147835 and a Leducq Transatlantic Network grant 16CVD01 to CKG, NIH grants HL123485 and HL147187 to CER. TAP was supported by NIH grant T32DK007044. MAH was supported by a Rubicon grant from the Netherlands Organization for Scientific Research and postdoctoral grants from the Amsterdam Cardiovascular Sciences Institute and the American Heart Association. LKS was supported by NIH grant T32HL007249-42 and American Heart Association grant 20PRE35200195.

## Additional information

### Funding

| Funder | Grant reference number | Author |
| --- | --- | --- |
| National Institutes of Health | DK091183 | Christopher K Glass |
| National Institutes of Health | HL147835 | Christopher K Glass |
| Leducq Transatlantic Network | 16CVD01 | Christopher K Glass |
| National Institutes of Health | T32DK007044 | Thomas A Prohaska |
| American Heart Association | postdoctoral grant | Marten A Hoeksema |
| Netherlands Organization for Scientific Research | Rubicon grant | Marten A Hoeksema |
| Amsterdam Cardiovascular Sciences Institute | postdoctoral grant | Marten A Hoeksema |
| National Institutes of Health | HL123485 | Casey E Romanoski |
| National Institutes of Health | HL147187 | Casey E Romanoski |
| National Institutes of Health | T32HL007249-42 | Lindsey K Stolze |
| American Heart Association | 20PRE35200195 | Lindsey K Stolze |

The funders had no role in study design, data collection and interpretation, or the decision to submit the work for publication.

### Author contributions

Zeyang Shen, Conceptualization, Data curation, Formal analysis, Software, Visualization, Writing – original draft, Writing – review and editing; Rick Z Li, Data curation, Formal analysis, Software, Writing – original draft, Writing – review and editing; Thomas A Prohaska, Investigation, Writing – original draft, Writing – review and editing; Marten A Hoeksema, Data curation, Investigation, Writing – original draft, Writing – review and editing; Nathan J Spann, Data curation, Investigation; Jenhan

Tao, Gregory J Fonseca, Conceptualization, Writing – original draft, Writing – review and editing; Thomas Le, Formal analysis; Lindsey K Stolze, Data curation, Resources; Mashito Sakai, Investigation, Resources, Supervision; Casey E Romanoski, Resources, Supervision; Christopher K Glass, Conceptualization, Funding acquisition, Project administration, Supervision, Writing – original draft, Writing – review and editing

### Author ORCIDs
Zeyang Shen ![ORCID] http://orcid.org/0000-0003-3870-1390
Rick Z Li ![ORCID] http://orcid.org/0000-0001-8518-5793
Marten A Hoeksema ![ORCID] http://orcid.org/0000-0001-5981-121X
Mashito Sakai ![ORCID] http://orcid.org/0000-0002-4908-2727
Casey E Romanoski ![ORCID] http://orcid.org/0000-0002-0149-225X
Christopher K Glass ![ORCID] http://orcid.org/0000-0003-4344-3592

### Ethics

Bone marrow cells were isolated from femurs and tibias of Cas9-expressing transgenic mice (Jackson Laboratory, No.028555) housed at the University of California San Diego animal facility on a 12-hour/12-hour light/dark cycle with free access to normal chow food and water. All of the mice were handled according to approved institutional animal care and use committee (IACUC) protocols (S01015) of the University of California San Diego to minimize pain and suffering.

### Decision letter and Author response

Decision letter https://doi.org/10.7554/eLife.70878.sa1
Author response https://doi.org/10.7554/eLife.70878.sa2

## Additional files

### Supplementary files

• Supplementary file 1. Tables include motif information for transcription factors (TFs), statistics of chromatin immunoprecipitation sequencing (ChIP-seq) peaks based on the ENCODE data, and the region targets and region-specific primers used for the CRISPR experiments.

• Transparent reporting form

### Data availability

All sequencing data generated during this study have been deposited in GEO under accession code GSE178080. The codes for data analysis and the processed files of the ENCODE data are available on our Github repository: https://github.com/zeyang-shen/spacing_pipeline. copy archived at swh:1:rev:e9a2a65795b8349843f1d2b7128395eb8e365015.

The following dataset was generated:

| Author(s) | Year | Dataset title | Dataset URL | Database and Identifier |
|---|---|---|---|---|
| Prohaska TA, Hoeksema MA, Spann NJ, Sakai M, Shen Z, Glass CK | 2021 | Systematic analysis of effects of naturally occurring insertions and deletions that alter transcription factor spacing identifies tolerant and sensitive transcription factor pairs | https://www.ncbi.nlm.nih.gov/geo/query/acc.cgi?acc=GSE178080 | NCBI Gene Expression Omnibus, GSE178080 |

*Continued on next page*

The following previously published datasets were used:

| Author(s) | Year | Dataset title | Dataset URL | Database and Identifier |
|---|---|---|---|---|
| Link VM, Glass CK | 2018 | Analysis of genetically diverse macrophages reveals local and domain-wide mechanisms that control transcription factor binding and function | https://www.ncbi.nlm.nih.gov/geo/query/acc.cgi?acc=GSE109965 | NCBI Gene Expression Omnibus, GSE109965 |
| Stolze LK, Romanoski CE | 2020 | Molecular Quantitative Trait Locus Mapping In Human Endothelial Cells Identifies Regulatory SNPs Underlying Gene Expression and Complex Disease Traits | https://www.ncbi.nlm.nih.gov/geo/query/acc.cgi?acc=GSE139377 | NCBI Gene Expression Omnibus, GSE139377 |

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
