## [Editor Report]

Transcription factors (TFs) bind to the DNA in a sequence-specific manner at TF binding sites (TFBSs) to control gene transcription. Hence, characterizing how TFs interact with DNA is key to uncover how gene regulation occurs and how this process can be disrupted in diseases. While the binding properties of a large portion of human TFs are well characterized, a remaining challenge lies in our knowledge of how TFs interact cooperatively at regulatory elements, either forming dimers or co-binding the same regions. In this manuscript, Shen et al. explored spacing patterns between TFBSs using previously published data sets and revealed that the dominant pattern is a relaxed range of spacing between collaborative factors and tolerance for InDels that change the TFBS spacing.

---

## [Decision Letter]

**Decision letter after peer review:**

[Editors’ note: the authors submitted for reconsideration following the decision after peer review. What follows is the decision letter after the first round of review.]

Thank you for submitting your work entitled "Natural genetic variation affecting transcription factor spacing at regulatory regions is generally well tolerated" for consideration by *eLife*. Your article has been reviewed by 2 peer reviewers, including Chris P Ponting as the Reviewing Editor and Reviewer #1, and the evaluation has been overseen by a Reviewing Editor and a Senior Editor.

Our decision has been reached after consultation between the reviewers. Based on these discussions and the individual reviews below, we regret to inform you that your work will not be considered further for publication in *eLife*.

The question under study without doubt is important and the natural experiment approach has several strengths. Nevertheless, both reviewers would have been more persuaded of its conclusions if large-scale experimental validations could have been reported and if these conclusions were generalized to other TFs and systems. A large number of methodological and statistical questions were also raised that together reduced the enthusiasm of the reviewers for publication. We hope that the comments provided below are helpful to you when you consider how and when to submit this work for publication.

*Reviewer #1:*

This manuscript by Shen et al. exploits pre-existing data (Link et al., 2018a; Keane et al., 2011) to explore whether InDels observed among 5 diverse mouse strains alter transcription factor binding affinity, using the PU.1 and C/EBPβ LDTFs as paradigm.

These wholly computational analyses provide evidence that affinities of "co-binding sites" correlate poorly with their spacing. Variants lying in PWMs alter affinity for their cognate factor, but also the co-binding factor, but the authors propose that there is little change in affinity resulting from spacing changes between them. This conclusion is drawn from indirect observations: (a) "co-binding" is never demonstrated, rather inferred from close proximity of ChIP-seq peaks, (b) changes in effect sizes (Table 1) following filtering with collaborative or unrelated factors, although whether these results are statistically robust requires further clarity.

The study would have benefited from the results drawn from the "natural experiments" being then validated experimentally, for example using reporter assays in cells whose TF spacing has been changed. Overall, its results were highly correlative, and provided little in the way of substantive novel observations.

Issues:

(i) line 201 "The remaining sites with InDels between PU.1 and C/EBPβ motifs, which should represent a clean set of spacing alterations, showed a diminished effect on TF binding (Table 1 "filtered by collabor. factors"; Figure 3D)." I'm unsure whether this is true, or else whether the change in significance (p-value Table 1) reflects the change in number of binding regions being considered. Similarly, the H3K27ac analysis referred to on line 232. Please comment on whether these effect size differences are significant.

(ii) Figure 5 does not provide new findings that are sufficient to merit having a Main Figure. Figure 5A. This analysis should be repeated with a more sensitive local aligner, e.g. LASTZ. Panel A simply reflects a technical effect (that of altered gap penalties) rather than any biological phenomenon (similarly Figure Supplements) and panels 5B,C illustrate one example, rather than a generalizable phenomenon.

*Reviewer #3:*

In this study, Shen et al. perform a meta-analysis of previously published data to unravel the effects of SNPs and InDels in the binding of CEBPB and PU.1 in macrophages. The authors find/confirm that alterations in the TF motif have an important impact on TF binding, while alterations on the spacing between the motifs does not have an effect on their binding. While this is an intriguing question, the study is rather straightforward, mostly confirms previous observations of the impact of PU.1 binding site alterations (although the presented analysis techniques are state-of-the-art and inspiring; and InDels are often not included, while here they are carefully assessed). The study could be enhanced with experimental validation (e.g. enhancer reporter assays); comparison with other computational techniques (e.g., deep learning); inclusion of additional data layers (e.g., gene expression); and expanding it to other TFs and systems to investigate whether these findings are generally true for other cell types. As it currently stands, the conclusions in the paper are a bit too general from the (biased) analyses performed.

1. Are LDTFs pioneer TFs and SDTFs non-pioneer TFs? We have only found this terminology in papers from this group.

2. The basis of the study are ChIP-seq peaks, but ChIP-seq peaks without the TF motif exist (sometimes a large fraction), can this be discussed? Particularly how false positive (i.e., indirect binding sites, or phantom peaks) impact the results.

3. Calculating spacing between motifs is a big challenge, because often a CRM contains multiple matches to the same TF. If TF1 has 3 matches, and TF2 has 4 matches, then there are a lot of distances between TF1 and TF2 motifs possible. The authors calculate only the distance between the two best scoring matches. Can this decision be justified by a thorough analysis? (e.g., do other distances between the best match of PU.1 and the 2nd best match of CEBP destroy or maintain the correlations?). In conclusion, a landscape overview of possible distances vis-a-vis genomic variation would be more informative than only using the distance between the two best matches.

4. Related – in Figure 1c. How is the distance between CEBPB and PU.1 motifs calculated on the singly PU.1 binding sites, do they also have CEBPB binding sites (and vice versa)? If yes, what are the differences between these regions and regions bound by CEBPB (e.g. is it because CEBPB motifs are weaker in these regions compared to regions bound by CEBPB)?

5. Also related – how much does spacing change with indels (e.g. from ~12bp, what is the final size distribution with indels)?

6. Figure Suppl 4 (Figure 2). In regards to the effect of the spacing, it seems that when the initial spacing between the motifs is around 60 bp, effects are weaker compared to when the initial size is between 20-40 and 80-100 bp; what could be the reason?

7. Figure 1d. Have sites with log reads CEBPB/PU.1 < 4 been filtered out?

8. Line 115 – the negative correlation between PU.1 and CEBP motifs 'implicates' synergistic binding and degenerative motifs, but this is speculation (it is not known which ChIP-seq events reflect true enhancers).

9. If CEBPB ChIP-seq signal is negatively correlated with the PU.1 motif score (Figure 1d), mutation of the PU.1 motif should increase CEBPB binding (Figure 2). Based on panel D, it seems that mutation of the PU.1 motif actually decreases CEBPB binding.

10. The quality of the SNPs and InDels are not discussed. InDels are more prone to false positives, this was not taken into account?

11. The quality of the ChIP-seq peaks is also not taken into account. Can a FRIP analysis be provided, for example plot FRIP versus the number of peaks across the samples.

12. PU.1 and CEBP were somehow chosen. An unbiased analysis starting from a larger motif collection would have been more interesting, to see if the properties of PU.1, CEBP, and their distance, emerges from the background of other motifs. Later on there is a JASPAR analysis – what were other high-scoring hits?

13. For the twelve collaborative factors predicted, some kind of (computational) validation would be useful.

14. For identifying cofactors exploiting other ChIP-seq data (Figure 3A), which ChIP-seq tracks were used and in which tissue? In general, data/code availability sections are missing.

[Editors’ note: further revisions were suggested prior to acceptance, as described below.]

Thank you for submitting your article "Systematic analysis of naturally occurring insertions and deletions that alter transcription factor spacing identifies tolerant and sensitive transcription factor pairs" for consideration by *eLife*. Your article has been reviewed by 3 peer reviewers, and the evaluation has been overseen by a Reviewing Editor and Jessica Tyler as the Senior Editor. The following individuals involved in review of your submission have agreed to reveal their identity: Jeff Vierstra (Reviewer #2).

The reviewers are in agreement that this paper is of potential interest for readers studying transcription factor (TF) binding site grammar/enhancer regulation. The work provides insight into the role of motif spacing alterations in TF binding/co-binding in the context of naturally occurring genetic variation. Overall, the data are properly analyzed and validated, although aspects of data analysis and presentation should be improved as outlined below, in a revised manuscript.

*Reviewer #1:*

Shen and colleagues investigated the role of motif spacing in regulating transcription factor (TF) binding or co-binding, specifically in the context of naturally occurring InDels in both human individuals and mouse strains, using previously published data (Hu et al., 2019; Link, Duttke, et al., 2018; Stolze et al., 2020).

The authors classified 75 TFs in K562 cells into constrained and relaxed spacing relationships with respect to their co-binding TFs, and found most of the TF pairs fall in the relaxed category. To illustrate whether spacing alterations affect TF binding and promoter-enhancer function, the authors analyzed the previously published data in mouse macrophages and human endothelial cells. They find that relaxed TF binding is highly tolerant to naturally occurring spacing alterations, which was further supported by CRISPR/Cas9 induced InDels in mouse macrophages.

The study would benefit from including other types of data available (i.e. gene expression and 3D interactions), to better examine the effects of spacing alterations and whether they are indeed related to promoter-enhancer functions.

1. The authors defined constrained and relaxed spacing relationships using public dataset including 75 TF ChIP-seq in K562 cells. K562 and HepG2 (analyzed in this study) are cancer cell lines. The karyotype of cancer cell lines, their own genetic variations and specific TF networks need to be carefully examined/ruled out before applying this to other data from healthy individuals.

2. Some TFs, especially the ones belong to the same families, share core motif sequences, and usually these TFs can co-localize to regulate gene expression. It is not clear how this case was handled in the pipeline.

3. (Figure 2A and 2B) The number of TF pairs considered/affected, the number of InDels at motifs/between motifs/in backgrounds, and the number of high-frequency/rare variants/singletons, need to be listed, which could further help illustrate the significance of any enrichment.

4. Line 165 – 168, "Since common variants are associated with less deleteriousness and rare variants with more deleteriousness (Lek et al., 2016), our data suggest that InDels between motifs of TFs with constrained spacing could be just as damaging as those at motifs whereas InDels between motifs of TFs with relaxed spacing might have a much weaker effect". This is speculation, and it would be better supported by at least some examples if not analysis/statistics, to show the deleterious effects. Probably this could be validated using CRISPR experiments as for relaxed ones.

5. The authors attempted to explore if these InDels eventually affect enhancer-promoter activity/function, but it's not clear whether enhancers and promoters were considered and whether they were considered separately in the analysis. Also, it would be great if the authors could assess whether spacing alterations investigated here in mouse macrophages affect gene expression, since RNA-seq/GRO-seq and PLAC-seq data are available from the published research. This information may help clarify the analysis, strengthen the conclusion and would be useful for readers interested in enhancer regulation.

*Reviewer #2:*

This paper attempts to address a long-standing question of how TFs collaborate to instantiate and maintain accessible and functional regulatory DNA. The authors make use of ENCODE data the investigate the extent TF spacing constraints in the genome and then integrate both human genetics data (gnomAD) and a compendium of ChIP-seq experiments performed in diverse mouse genetic backgrounds to test whether motif spacing has a significant effect on TF binding. While I appreciated the authors utilization of many datasets to test for spacing effects (large ENCODE data to identify motif pairs with spacing constraints, human genetic to look for signals of negative selection of natural variation effecting spacing and mouse TF binding data to 'test' for spacing effects), I find that computational analysis within this paper is quite shallow, leading to mostly obvious conclusion that variation within the TF-DNA interaction interface are critical for TF binding. Furthermore, While the editing experiments are a nice addition in the revision, I am not sure that they provide much in the way of validating or generalizing their claims. Finally, the authors should more thoroughly place their work in the context of previous studies.

1. The constrained vs. relaxed spacing analysis has a high likelihood to be confounded by latent genome architecture. Specific class of retrotransposable elements are known to 'template' regulatory DNA (see Bourque, G. et al. 2008 Genome Res. (10.1101/gr.139105.112), Kunarso et al. 2010 Nat. Genetics (10.1038/ng.600), and an analysis of co-binding/spacing constraints from ENCODE data: Wang et al., 2012 Genome Res.(10.1101/gr.139105.112)). The authors should perform an analysis that accounts for repetitive DNA that encode competent cis-regulatory DNA elements that have been templated across the genome. At a minimum these previous works should be cited.

2. The authors should comment on how rigid DNA-encoded TF spacing is not supported by evolutionary studies, which have shown an excess of TF motif turnover within regulatory DNA (see work from Duncan Odom's group, Vierstra el., 2014 Science for a direct mouse-human comparison).

3. While Leveraging CRISR/Cas9 editing to generate a broad spectrum of 'spacing' alleles is a clever approach to tackle the experimentally test the effect of motif spacing on TF (co-)binding, the experiment is very underpowered to test the generalizability of the authors claims. Did the authors select single cell clones from the editing experiment or just look at bulk edited populations? If the former, it is unclear how any conclusions can be made from a mixture of edited cells that have a spectrum of indels (also likely carrying two different alleles).

*Reviewer #3:*

Transcription factors (TFs) bind to the DNA in a sequence-specific manner at TF binding sites (TFBSs) to control gene transcription. Hence, characterizing how TFs interact with DNA is key to uncover how gene regulation occurs and how this process can be disrupted in diseases. While the binding properties of a large portion of human TFs are well characterized, a remaining challenge lies in our knowledge of how TFs interact cooperatively at regulatory elements, either forming dimers or co-binding the same regions. In this manuscript, Shen et al. explored spacing patterns between TFBSs. Relying on ChIP-seq data, they developed a new methodology to predict TF pairs harbouring constrained or relaxed spacing patterns between their TFBSs. The authors made their code available, which allows reproducibility and exploration; this should be a requirement in the field but is not always complied with so we thank the authors for this. When applied to a limited set of TFs with ChIP-seq data in K562 cells, the authors predicted that TF pairs primarily bind to DNA with relaxed spacing between their TFBSs. Nevertheless, they were able to highlight already known as well as novel specific pairs of TF harboring constrained spacing. Next, the authors leveraged naturally occurring small insertions and deletions in the human population and mouse strains to confirm that altering spacing between TFBSs of TF pairs with relaxed spacing patterns has limited effect. This observation was further supported by synthetic spacing alterations induced by CRISPR-Cas9 experiments. The study is overall well designed and addresses an important challenge in our understanding of TF-DNA interactions and TF cooperation.

Nevertheless, we believe that there are some methodological limitations that favor the identification of relaxed spacing patterns, which should be better outlined in the manuscript to allow the reader to fully comprehend the results. From the title and first sections of the manuscript the readers are given the impression that relaxed and constrained spacing instances are about to be described and analysed with an equal importance. However, more focus is given to the relaxed spacing with both the mouse and CRISPR analyses exclusively dedicated to this with no clear explanation why. It would be useful to the readers to have this explicitly outlined by the authors. Finally, the terminology associated with TF-DNA interactions is very often incorrect, which confuses the readers and should be addressed. Please see below for our detailed comments.

1. The terminology associated with TF binding events is inappropriate. The authors use "ChIP-seq peaks", "TFBSs", and "motifs" almost interchangeably, which is not correct. The inconsistency in the terminology makes it difficult to fully comprehend what the authors meant/did.

An example is one of the first sentences in the Introduction: "TFs bind to short, degenerate sequences at promoters and enhancers, often referred to as TF binding motifs." The sequences bound by TFs in promoters and enhancers are TFBSs while TF binding motifs are computational representations of TFBS sets, which can be represented in many ways such as consensus motifs, PFMs, etc.

The next sentence claims that "TFs bind in an inter-dependent manner to closely spaced motifs." Motifs cannot be closely spaced but TFBSs are. Another example is the subsection "Motif identification" in the Methods section while the authors describe the prediction of TFBSs (using motifs).

2. More details should be provided to the Methods section. We acknowledge that the authors provide their code for inspection but outlining all methodological details in the manuscript would help in the clear understanding of the methods used. For instance:

a. The authors do not describe how they selected de novo motifs using HOMER (only best motif?, any p-value threshold?, any specific background used?)

b. For the TFBS predictions, the authors used a FPR threshold of 0.1%, but which was the specific tool that they used for that? FPR computation depends on background expectation, what was used (e.g. 25% A, C, G, or T or nucleotide composition of the genome)?

c. P. 23, lines 466-470. The authors described that they conducted a permutation test but then described that the null distribution was obtained using random spacing values between 0 and 100. If the null distribution is obtained by randomly selecting values between 0 and 100, it does not correspond to a permutation. A permutation test would imply permuting the observed spacing values.

d. P. 25, lines 513-516. It is not clear to us why the authors considered subsets of mutations when overlapping TFBSs predicted for the ChIP'ed TFs (only if 2-bit difference in motif score) but not for mutations overlapping TFBSs predicted by MAGGIE (all mutations). Why not consider all mutations in all cases?

3. The methodology used has several limitations that are not described by the authors. We encourage the authors to clearly outline them to the readers. Furthermore, we believe that these limitations favor the identification of relaxed spacing, which should be acknowledged, especially since the majority of the work focuses on alterations of spacing for TF pairs with relaxed spacing patterns.

a. It is well documented that TF binding preferences for TFs binding in close proximity (e.g. as dimers) can be altered. For instance, Jolma et al. (https://www.nature.com/articles/nature15518) used CAP-SELEX to reveal that "Most TF pair sites identified involved a large overlap between individual TF recognition motifs, and resulted in recognition of composite sites that were markedly different from the individual TF's motifs." As the authors relied on TF binding profiles (or motifs) corresponding to the binding preferences of TFs binding as monomers, it is possible that they will miss cooperative binding inducing a change in binding preferences. Furthermore, the authors did not consider overlapping motifs, which is again precluding the identification of constrained spacing.

b. The authors rely on ChIP-seq data to identify TF cooperation. While this is fine overall, this data does not allow the authors to know whether two identified TFs bind to their TFBSs on the same molecules. Indeed, ChIP-seq being a bulk experiment, it does not allow to discriminate between true co-occupancy on the same molecule or not. This should be discussed to put the work in a larger context to the readers. See https://www.cell.com/molecular-cell/pdf/S1097-2765(20)30793-0.pdf for a reference.

c. The de novo motif enrichment does not ensure that the motif found is actually the one bound by the ChIP'ed TF. Indeed, the motifs found for ARID1B and ARID2 correspond to GATA motifs while ARID TFs bind to more general A+T rich motifs. It is unclear whether the signal observed for these TFs is due to their direct interaction with DNA.

4. It would have been nice to perform similar experiments as the mouse and CRISPR ones but considering TF pairs with fixed spacing of their TFBSs. If the authors decide not to, they should at least clearly state to the readers that they more specifically focused on relaxed spacing and why.

5. It would have been interesting to provide information about the prevalence of cobinding events in the different ChIP-seq datasets. For instance, what is the percentage of ChIP-seq peaks that contain a predicted TFBS for each TF? What is the percentage of such peaks that contain cobinding events? Is there a difference in numbers b/w constrained or relaxed spacing. Providing this information would help the readers to put these observations into a more general context of TF binding regions.

6. It would be nice to have similar plots as in Figure 3B for pairs of TFs with constrained spacing to show how it contrasts.

7. In all analyses, it seems that more deletions than insertions were observed (see Figure 2A for instance). It would be interesting to see if the results recapitulate when considering insertions and deletions independently.

8. It is unclear to us what is the analysis of MAGGIE-predicted TFBSs adding to the story. Moreover, the authors claim that MAGGIE identifies "functional" TFBSs but the authors do not provide any specific evidence of function (and which function?) in our opinion.

9. It seems that there is a periodic pattern in Figure 5E with log-odds ratio periodically equal to 0 (at least with indel sizes < -45). Could this correspond to the minor groove width periodicity of ~10bp (see for instance https://www.cell.com/cell/pdf/S0092-8674(18)31312-6.pdf for periodicity of mutations)? Could the authors comment on that?

10. In several figures, the authors should provide all points instead of summarizing with boxplots, which are frowned upon as they hide data distribution.

11. P. 9 line 178: the authors mentioned 50 millions SNPs but we do not find where this data is used as observing SNPs would not alter spacing.

12. P. 2 line 46: "implicating their effect…" we would rather write "suggesting their effect…".

13. The authors seem to have ignored homodimers. Maybe their methodology could be extended to consider spacing b/w TFBSs for the same TF.

14. TF naming is sometimes inconsistent in the text and figures. For example, SPI1 and PU.1 are both used in Figure 3. Similarly, we recommend revisiting the text for p65 and RELA, as well as C/EBPβ and CEBPB.

15. It is not clear where the authors retrieved HepG2 cell line data from (used in the Figure 1 —figure supplement 4). Was this data processed in the same way as K562 cell line data?

16. P. 8 line 152: we suggest replacing "we overlaid" with "we mapped".

17. P. 1 line 30: "ChIP-sequencing" should be replaced with "ChIP-seq" for consistency.

18. Figure 3D,F,H : instead of motif score this should be TFBS score.

---

## [Author Response]

[Editors’ note: the authors resubmitted a revised version of the paper for consideration. What follows is the authors’ response to the first round of review.]

Reviewer #1:This manuscript by Shen et al. exploits pre-existing data (Link et al., 2018a; Keane et al., 2011) to explore whether InDels observed among 5 diverse mouse strains alter transcription factor binding affinity, using the PU.1 and C/EBPβ LDTFs as paradigm.These wholly computational analyses provide evidence that affinities of "co-binding sites" correlate poorly with their spacing. Variants lying in PWMs alter affinity for their cognate factor, but also the co-binding factor, but the authors propose that there is little change in affinity resulting from spacing changes between them. This conclusion is drawn from indirect observations: (a) "co-binding" is never demonstrated, rather inferred from close proximity of ChIP-seq peaks, (b) changes in effect sizes (Table 1) following filtering with collaborative or unrelated factors, although whether these results are statistically robust requires further clarity.The study would have benefited from the results drawn from the "natural experiments" being then validated experimentally, for example using reporter assays in cells whose TF spacing has been changed. Overall, its results were highly correlative, and provided little in the way of substantive novel observations.

We thank Reviewer #1 for these comments, which led us to perform additional experiments and analyses. To examine the robustness of our conclusion in another system, we extended the findings in mouse macrophages to investigate the spacing relationships of ERG and p65 in a previously studied cohort of human endothelial cells. These studies strongly reinforced the findings in macrophages with respect to tolerance of ERG and p65 binding on intervening InDels (New Figure 4). The major general criticism of Reviewer #1 is lack of experimental validation. Rather than using reporter assays which may not reflect natural chromatin environments, we performed new experiments in which we used CRISPR/Cas9 methodology to introduce a range of InDels between endogenous PU.1 and C/EBPβ binding sites at 6 different genomic locations (three containing natural InDels and three without) and measured the effects of different InDels altering spacing on transcription factor binding. To our knowledge, this is the first application of CRISPR technology for this purpose. These studies demonstrated remarkable stability of C/EBPβ binding at deletions up to the point of destruction of either the CEBP or PU.1 motif, strongly supporting the original conclusions of our manuscript (New Figure 5).

Issues:(i) line 201 "The remaining sites with InDels between PU.1 and C/EBPβ motifs, which should represent a clean set of spacing alterations, showed a diminished effect on TF binding (Table 1 "filtered by collabor. factors"; Figure 3D)." I'm unsure whether this is true, or else whether the change in significance (p-value Table 1) reflects the change in number of binding regions being considered. Similarly, the H3K27ac analysis referred to on line 232. Please comment on whether these effect size differences are significant.

We thank Reviewer 1 for this comment. Since the original submission, we made changes to our categorization method for binding regions and were able to distinguish genetic variations affecting PU.1, CEBP, or other functional motifs identified by our recently developed method MAGGIE (Shen et al., 2020) and those altering spacing without affecting motifs. The statistical significances or effect sizes are reported in the new Figure 3 and new Figure 4.

(ii) Figure 5 does not provide new findings that are sufficient to merit having a Main Figure. Figure 5A. This analysis should be repeated with a more sensitive local aligner, e.g. LASTZ. Panel A simply reflects a technical effect (that of altered gap penalties) rather than any biological phenomenon (similarly Figure Supplements) and panels 5B,C illustrate one example, rather than a generalizable phenomenon.

We accepted reviewer’s suggestion on the original Figure 5 as not being sufficient to merit a main figure and excluded these findings in the new submission. Instead, we leveraged more than 60 million InDels from gnomAD data, which were based on more than 75,000 genomes from unrelated individuals, to investigate the potential for selective pressure against InDels altering transcription factor spacing. We found that the InDels between motifs of TFs with a relaxed spacing relationship are associated with less deleteriousness, whereas InDels that occur between motifs of constrained TFs are enriched with singletons, associated with stronger deleterious effects (New Figure 2).

Reviewer #3:In this study, Shen et al. perform a meta-analysis of previously published data to unravel the effects of SNPs and InDels in the binding of CEBPB and PU.1 in macrophages. The authors find/confirm that alterations in the TF motif have an important impact on TF binding, while alterations on the spacing between the motifs does not have an effect on their binding. While this is an intriguing question, the study is rather straightforward, mostly confirms previous observations of the impact of PU.1 binding site alterations (although the presented analysis techniques are state-of-the-art and inspiring; and InDels are often not included, while here they are carefully assessed). The study could be enhanced with experimental validation (e.g. enhancer reporter assays); comparison with other computational techniques (e.g., deep learning); inclusion of additional data layers (e.g., gene expression); and expanding it to other TFs and systems to investigate whether these findings are generally true for other cell types. As it currently stands, the conclusions in the paper are a bit too general from the (biased) analyses performed.

We thank Reviewer #3 for these recommendations. To generalize our findings from mouse macrophages, we conducted four additional lines of analyses:

First, we developed a new analytical pipeline for determining whether the spacing relationships between transcription factor pairs are relaxed (relatively independent of spacing) or constrained (presence of a fixed spacing interval) and applied this method to ChIP-seq data sets for 75 transcription factors determined in K562 cells by the ENCODE consortium. Nearly all of the factors examined primarily exhibited relaxed spacing relationships with the majority of co-bound factors, consistent with the major conclusions of our original manuscript. However, about half of these factors also exhibited constrained relationships with specific TF partners. We confirmed previously identified constrained relationships, such as for GATA1 and TAL1 and identified several previously unrecognized constrained relationships (New Figure 1).

Second, as noted in our response to Reviewer #1, having defined the pairwise spacing relationships for these 75 factors, we then leveraged more than 60 million InDels from gnomAD data, which were based on more than 75,000 genomes from unrelated individuals, to investigate the potential for selective pressure against InDels that occur at or between motifs. We found that the InDels between motifs of TFs with a relaxed spacing relationship are associated with less deleteriousness, whereas InDels that occur between motifs of constrained TFs are enriched with singletons, associated with stronger deleterious effects (New Figure 2).

Third, as noted in our response to Reviewer #1, we extended the findings in mouse macrophages to investigate the spacing relationships of ERG and p65 in a previously studied cohort of human endothelial cells, in which it was possible to consider InDels as quantitative trait loci with respect to the binding of ERG and p65. These studies strongly reinforced the findings in macrophages with respect to tolerance of ERG and p65 binding on intervening InDels (New Figure 4).

Lastly, as noted in our response to Reviewer #1, we performed new experiments in which we used CRISPR/Cas9 methodology to introduce a range of InDels into six endogenous genomic loci of PU.1 and C/EBPβ co-binding in macrophages. This mutagenesis approach resulted in populations of cells containing InDels ranging from +5 bp (insertions) to >-30 bp (deletions) between the PU.1 and C/EBP binding sites. We then performed ChIP for C/EBPβ in control cells and in CRISPR/Cas9 mutated cells and deeply sequenced the immunoprecipitated DNA for the targeted locations using paired end sequencing that enabled determination of the InDel size. By comparing the ChIP tag counts in control cells to the ChIP tag counts at each interval of insertion or deletion, we could calculate an odds ratio to quantify the effect of the InDel on C/EBPβ binding. These studies demonstrated remarkable stability of C/EBPβ binding at deletions up to the point of destruction of either the CEBP or PU.1 motif, strongly supporting the original conclusions of our manuscript (New Figure 5).

1. Are LDTFs pioneer TFs and SDTFs non-pioneer TFs? We have only found this terminology in papers from this group.

Our interpretation of the term pioneer factor is that it is a sequence-specific transcription factor that has the ability to interact with its DNA recognition motif in the context of closed chromatin and initiate chromatin remodeling. This is the essential first step required for selection of new enhancers that drive transitions in cell fate. Many LDTFs have this capability, such as PU.1. Importantly, the ability of pioneer factors to interact with DNA recognition motifs in closed chromatin is necessary but not sufficient for enhancer selection and activation. From our studies of macrophages, for example, the binding of pioneer factors such as PU.1 always requires mutually dependent interactions with additional transcription factors, which we define as collaborative binding. We use the term SDTF to refer to a transcription factor that is acutely responding to an internal or external signal. It may or may not have properties of a pioneer factor or LDTF. As these details of the terminology are not central to the main message of the manuscript, we did not provide extensive explanations to their definitions, but can expand if the reviewer recommends that this would be helpful for broader accessibility.

2. The basis of the study are ChIP-seq peaks, but ChIP-seq peaks without the TF motif exist (sometimes a large fraction), can this be discussed? Particularly how false positive (i.e., indirect binding sites, or phantom peaks) impact the results.

We agree with this concern. Based on reviewer’s suggestion, our analyses in the new submission involving ChIP-seq peaks all focused on those with highly confident TF motifs to minimize the impacts of false positives or indirect binding sites on our conclusions. Confident motifs all had a PWM score passing false positive rate (FPR) < 0.1% and a location within 50 bp from the peak centers. We found that these two criteria were important for recovering true spacing relationships from TF binding sites based on our examination of known constrained spacing of GATA1 and TAL1 (new Figure 1-figure supplement 1).

3. Calculating spacing between motifs is a big challenge, because often a CRM contains multiple matches to the same TF. If TF1 has 3 matches, and TF2 has 4 matches, then there are a lot of distances between TF1 and TF2 motifs possible. The authors calculate only the distance between the two best scoring matches. Can this decision be justified by a thorough analysis? (e.g., do other distances between the best match of PU.1 and the 2nd best match of CEBP destroy or maintain the correlations?). In conclusion, a landscape overview of possible distances vis-a-vis genomic variation would be more informative than only using the distance between the two best matches.

We thank reviewer’s comment for inspiring us to improve our computational pipeline of characterizing spacing relationships for TF pairs. Based on our examination of GATA1 and TAL1 (new Figure 1—figure supplement 1), we found that best motifs could generally reflect true spacing relationship, but considering all the high-confidence motifs (FPR < 0.1%) could recover an even stronger signal. In addition, restricting the relative locations of motifs from the ChIP-seq peak centers could substantially reduce false positives. Therefore, in our pipeline of characterizing spacing relationships, we computed all possible pairs of confident TF motifs located within 50 bp from the peak centers. However, considering multiple matches would be trickier for genetic variation analysis. For example, one region can have an InDel altering spacing between two best matches but not altering spacing between the best match of PU.1 and the second best match of C/EBP. As the reviewer pointed out, since it can be a challenge to categorize genetic variation when considering multiple matches at the same time, we categorized genetic variation only based on the best match motifs. To examine the reproducibility of our conclusions, we did four independent comparisons between mouse strains and observed very similar effect size relationships of spacing alterations and motif mutations. These findings were further supported by analyses in human endothelial cells and experimental validations we included in the new submission.

4. Related – in Figure 1c. How is the distance between CEBPB and PU.1 motifs calculated on the singly PU.1 binding sites, do they also have CEBPB binding sites (and vice versa)? If yes, what are the differences between these regions and regions bound by CEBPB (e.g. is it because CEBPB motifs are weaker in these regions compared to regions bound by CEBPB)?

We thank Reviewer #3 for this comment. The spacing plots of singly PU.1 binding sites are not included in the revised manuscript due to the addition of many other experimental results and analyses that provide more substantial evidence for the original conclusions.

5. Also related – how much does spacing change with indels (e.g. from ~12bp, what is the final size distribution with indels)?

We included the size distribution of InDels for every dataset we analyzed in the new submission as shown in new Figure 2A, new Figure 3—figure supplement 1, new Figure 4—figure supplement 3, and new Figure 5B. Overall, the majority of naturally occurring InDels are less than 5 bp in length, and synthetic InDels generated by CRISPR/Cas9 system provides a wider range with deletions as large as >30 bp.

6. Figure Suppl 4 (Figure 2). In regards to the effect of the spacing, it seems that when the initial spacing between the motifs is around 60 bp, effects are weaker compared to when the initial size is between 20-40 and 80-100 bp; what could be the reason?

We thank Reviewer #3 for this comment. After re-doing the analysis with improved computational pipeline and aggregating the results from all the four pairwise comparisons of mouse strains, we no longer saw a weaker effect around 60 bp. This result is presented in new Figure 3-Supplement 3.

7. Figure 1d. Have sites with log reads CEBPB/PU.1 < 4 been filtered out?

Part of original Figure 1D is now shown as Figure 3B in the new submission. The regions used to plot new Figure 3B are the same ones used in new Figure 3A. These regions were not explicitly filtered by tag counts but by confident PU.1 and CEBP motifs. Most of these regions turned out to have more than 16 ChIP-seq tags, showing strong TF binding activity.

8. Line 115 – the negative correlation between PU.1 and CEBP motifs 'implicates' synergistic binding and degenerative motifs, but this is speculation (it is not known which ChIP-seq events reflect true enhancers).

Since this comment and comment #9 both refer to Figure 1D of the original manuscript, comment #8 will be responded together with comment #9 below.

9. If CEBPB ChIP-seq signal is negatively correlated with the PU.1 motif score (Figure 1d), mutation of the PU.1 motif should increase CEBPB binding (Figure 2). Based on panel D, it seems that mutation of the PU.1 motif actually decreases CEBPB binding.

The original Figure 1D is no longer included in the revised manuscript due to the addition of many other experimental results and analyses that strengthen the original conclusions. Although the p value was significant in the original Figure 1D, the Spearman correlation coefficient of -0.1 indicates almost no correlation of C/EBP binding with the PU.1 motif score. Importantly, this is not the same as an effect of a mutation in the PU.1 motif on C/EBP binding. Many independent lines of evidence, including evidence in the present study, indicate that mutation of a PU.1 motif (over a range of motif scores) results in a decrease in collaborative binding of C/EBP.

10. The quality of the SNPs and InDels are not discussed. InDels are more prone to false positives, this was not taken into account?

The InDels from mouse strains and humans in this study have all passed quality filters according to the source papers. For instance, according to the source paper for the genetic variation of inbred mouse strains (Keane et al., 2011), the false positive error per 10 kb for InDels is only 0.09, which is too low to bias our conclusions.

11. The quality of the ChIP-seq peaks is also not taken into account. Can a FRIP analysis be provided, for example plot FRIP versus the number of peaks across the samples.

All ChIP-seq peaks used in the new submission were identified significantly reproducible (IDR < 0.05) based on biological replicates from ENCODE data portal and source papers. The statistics of ChIP-seq data generated through this study are included in Figure 5-table supplement 2. All the ENCODE data meet the quality metrics described in Landt et al., 2012, including FRIP > 1%. All the mouse macrophage ChIPseq data analyzed in new Figure 3 also have FRIP greatly exceeding 1% with the minimum FRIP being 3.65%. Author response image 1 summarizes the FRIP in relationship with peak number across PU.1 and C/EBPβ samples of five mouse strains.

**Author response image 1. sa2fig1:** 

12. PU.1 and CEBP were somehow chosen. An unbiased analysis starting from a larger motif collection would have been more interesting, to see if the properties of PU.1, CEBP, and their distance, emerges from the background of other motifs. Later on there is a JASPAR analysis – what were other high-scoring hits?

We included an unbiased systematic analysis of a large collection of TFs in the new submission (new Figure 1). We analyzed 75 transcription factors determined in K562 cells by the ENCODE consortium to obtain a global view of the spacing relationships among TFs.

13. For the twelve collaborative factors predicted, some kind of (computational) validation would be useful.

Different from our original way of identifying collaborative factors based on spacing relationships, in the new submission we applied our recently developed method MAGGIE (Shen et al., 2020) to identify a set of functional motifs. This method associates motif changes due to genetic variation and TF binding changes, and the motifs with significant associations are likely to be collaborative factors as validated in our MAGGIE paper.

14. For identifying cofactors exploiting other ChIP-seq data (Figure 3A), which ChIP-seq tracks were used and in which tissue? In general, data/code availability sections are missing.

In the revised manuscript, we transitioned to using our recently developed MAGGIE method to identify cofactors/collaborative factors. Data/code availability sections are now provided via the MAGGIE paper.

[Editors’ note: what follows is the authors’ response to the second round of review.]

Reviewer #1:Shen and colleagues investigated the role of motif spacing in regulating transcription factor (TF) binding or co-binding, specifically in the context of naturally occurring InDels in both human individuals and mouse strains, using previously published data (Hu et al., 2019; Link, Duttke, et al., 2018; Stolze et al., 2020).The authors classified 75 TFs in K562 cells into constrained and relaxed spacing relationships with respect to their co-binding TFs, and found most of the TF pairs fall in the relaxed category. To illustrate whether spacing alterations affect TF binding and promoter-enhancer function, the authors analyzed the previously published data in mouse macrophages and human endothelial cells. They find that relaxed TF binding is highly tolerant to naturally occurring spacing alterations, which was further supported by CRISPR/Cas9 induced InDels in mouse macrophages.The study would benefit from including other types of data available (i.e. gene expression and 3D interactions), to better examine the effects of spacing alterations and whether they are indeed related to promoter-enhancer functions.

We thank Reviewer #1 for these comments and the constructive recommendations. We now clarify in the revised manuscript that the major objectives of these studies are to determine the effects of spacing alterations on the local binding and function of TFs. We agree that placing these studies in the context of promoter-enhancer functions would be of interest. This is a question that we previously investigated in the broader context of SNPs and InDels that alter TF binding by directly altering TF binding sites (Heinz et al., 2013, Link et al., 2018, Hoeksema et al., 2021). From this prior work, we found that establishing direct relationships between TF binding at specific enhancers and target genes is often confounded by enhancer redundancy and the presence of co-occurring genetic variants of uncertain significance. Although many examples can be identified in which a motif mutation impairs TF binding at an enhancer and results in corresponding downregulation of a PLAC-seq connected gene, there are also many examples where this does not occur. At present, the ability to predict whether a variant-induced change in TF binding results in a change in gene expression is limited. Because of this limitation, we focused these studies on local consequences of TF activity. The prior version of the manuscript utilized H3K27ac as a readout based on the principle that TFs act to recruit co-activators with histone acetyltransferase activity. As suggested, we now include additional analyses of altered spacing on nascent transcription provided by GRO-seq data (see response to Comment 5), which more directly assesses transcriptional activity. These new analyses also document that relaxed spacing requirements for DNA binding are tolerated with respect to local transcriptional activity. We think that this result is interesting and significant because it extends the concept of spatial tolerance to the entire ensemble of factors that must be assembled to mediate transcriptional initiation. We thank Reviewer #1 for recommending this additional line of investigation, which is further described in response to Comment 5.

1. The authors defined constrained and relaxed spacing relationships using public dataset including 75 TF ChIP-seq in K562 cells. K562 and HepG2 (analyzed in this study) are cancer cell lines. The karyotype of cancer cell lines, their own genetic variations and specific TF networks need to be carefully examined/ruled out before applying this to other data from healthy individuals.

We thank Reviewer #1 for raising the potential for confounding effects of karyotypic variations of K562 cells and potentially K562-specific TF ChIP-seq peaks when overlaying these peaks with genetic variations in the population. To evaluate the extent to which K562-specific peaks could affect our results, we downloaded SNVs and indels of K562 cells (Zhou et al., 2019) and overlayed them with our TF ChIP-seq peaks called from ENCODE data. We found only ~19% of co-binding peaks contain at least one K562 SNV or InDel (proportions shown in Author response image 2), and we observed similar proportions in relaxed and constrained TF pairs. Importantly, Zhou et al. found 98% SNVs and 79% InDels of K562 cells overlapped with those in the general population based on dbSNP. Therefore, our conclusion of transcription factor binding is largely based on regions with no K562 variants or variants that are common in the general population, and the K562-specific variants only accounted for a very small fraction in our analyses.

2. Some TFs, especially the ones belong to the same families, share core motif sequences, and usually these TFs can co-localize to regulate gene expression. It is not clear how this case was handled in the pipeline.

We thank Reviewer #1 for raising this significant point. We agree that TFs that belong to a conserved family and recognize the same motif require special consideration. For example, JUN and JUND are both AP-1 factors and recognize very similar motifs. If JUN and JUND ChIP-seq peaks don’t overlap at all, there is no way for their motif spacing to be within the range of +/- 100 bp. Therefore, we only calculated spacing for overlapping peaks (i.e., co-binding peaks). For an overlapping peak (200-300 bp), there can be two scenarios: JUN and JUND binding sites (10bp sequence matching their motifs) overlap, or they don’t. Since we identified the binding sites for each of the two TFs from their corresponding 200-bp peaks by restricting the distance to peak centers, JUN and JUND binding sites aren’t necessarily at the same position. From this analysis, we found that a lot of JUN and JUND binding sites don’t overlap. We computed spacing for those non-overlapping binding sites and plotted the spacing relationship. This analysis indicated that the JUN and JUND with these characteristics have a significant relaxed spacing relationship. To better visualize the spacing relationships between TFs of the same family, we decreased the heights of connections within the same family in revised Figure 1C. We discussed this specific problem in the Results section (line 131 under “No Markup” view) and clarified how we handled TFs with shared core motif sequences in the Methods section under “TF binding site identification”.

3. (Figure 2A and 2B) The number of TF pairs considered/affected, the number of InDels at motifs/between motifs/in backgrounds, and the number of high-frequency/rare variants/singletons, need to be listed, which could further help illustrate the significance of any enrichment.

We included these numbers as new Figure 2-table supplement 1.

4. Line 165 – 168, "Since common variants are associated with less deleteriousness and rare variants with more deleteriousness (Lek et al., 2016), our data suggest that InDels between motifs of TFs with constrained spacing could be just as damaging as those at motifs whereas InDels between motifs of TFs with relaxed spacing might have a much weaker effect". This is speculation, and it would be better supported by at least some examples if not analysis/statistics, to show the deleterious effects. Probably this could be validated using CRISPR experiments as for relaxed ones.

Previous studies (Ng et al., 2014) have validated the deleterious effects of InDels between TF binding sites of constrained TFs, so we included citations of these studies at the end of section “Natural genetic variants altering spacing between relaxed transcription factors are associated with less deleteriousness in human populations” in the Results section.

5. The authors attempted to explore if these InDels eventually affect enhancer-promoter activity/function, but it's not clear whether enhancers and promoters were considered and whether they were considered separately in the analysis. Also, it would be great if the authors could assess whether spacing alterations investigated here in mouse macrophages affect gene expression, since RNA-seq/GRO-seq and PLAC-seq data are available from the published research. This information may help clarify the analysis, strengthen the conclusion and would be useful for readers interested in enhancer regulation.

We thank Reviewer #1 for these suggestions, as noted in our response to the general comments above. Our original analyses considered all TF binding sites at promoters and enhancers. In response to this request, we conducted additional analysis to examine the effects of InDels on promoters and enhancers separately. Most of the informative genetic variants are located at enhancers, and relatively few are within promoters. Despite that, we saw consistent relationships in promoters and enhancers, as exemplified by the C/EBPβ ChIP-seq results, which are now included as Figure 3—figure supplement 4.

Reviewer #2:This paper attempts to address a long-standing question of how TFs collaborate to instantiate and maintain accessible and functional regulatory DNA. The authors make use of ENCODE data the investigate the extent TF spacing constraints in the genome and then integrate both human genetics data (gnomAD) and a compendium of ChIP-seq experiments performed in diverse mouse genetic backgrounds to test whether motif spacing has a significant effect on TF binding. While I appreciated the authors utilization of many datasets to test for spacing effects (large ENCODE data to identify motif pairs with spacing constraints, human genetic to look for signals of negative selection of natural variation effecting spacing and mouse TF binding data to 'test' for spacing effects), I find that computational analysis within this paper is quite shallow, leading to mostly obvious conclusion that variation within the TF-DNA interaction interface are critical for TF binding. Furthermore, While the editing experiments are a nice addition in the revision, I am not sure that they provide much in the way of validating or generalizing their claims. Finally, the authors should more thoroughly place their work in the context of previous studies.

We thank Dr. Vierstra for highlighting aspects of the manuscript requiring further clarification and suggestions for better placing these studies in the context of prior work.

1. The constrained vs. relaxed spacing analysis has a high likelihood to be confounded by latent genome architecture. Specific class of retrotransposable elements are known to 'template' regulatory DNA (see Bourque, G. et al. 2008 Genome Res. (10.1101/gr.139105.112), Kunarso et al. 2010 Nat. Genetics (10.1038/ng.600), and an analysis of co-binding/spacing constraints from ENCODE data: Wang et al., 2012 Genome Res.(10.1101/gr.139105.112)). The authors should perform an analysis that accounts for repetitive DNA that encode competent cis-regulatory DNA elements that have been templated across the genome. At a minimum these previous works should be cited.

We thank Dr. Vierstra for bringing to our attention the possibility that specific classes of retrotransposable elements could be a confounding factor. To address this concern, we performed additional analyses to account for repetitive DNA regions and cited the suggested previous works. The results have been included in new Figure 1—figure supplement 6, Figure 1figure supplement 7, and Figure 1-table supplement 3. Codes newly generated for this analysis have been uploaded to our Github repository. Specifically, since DNA repetitive regions such as transposable elements are known to harbor TF binding sites and specific TF co-binding (Bourque et al., 2008; Kunarso et al., 2010), we further examined whether the spacing relationships of TFs could be different in repetitive and nonrepetitive regions. To study this, we applied the same pipeline to the subsets of TF ChIP-seq peaks in repetitive and nonrepetitive regions. As a result, most of the relaxed spacing relationships remained regardless of repetitive or nonrepetitive regions (Figure 1—figure supplement 6). Some constrained TF pairs, however, showed constrained spacing only in repetitive regions and not in non-repetitive regions (Figure 1-table supplement 3). For example, EGR1 and JUND exhibited a constrained spacing at 29 bp (Figure 1D), but this relationship is observed specifically in SINEs (Figure 1—figure supplement 7). Such observation is consistent with previous studies that discovered specific motif pairs in repetitive regions (J. Wang et al., 2012).

2. The authors should comment on how rigid DNA-encoded TF spacing is not supported by evolutionary studies, which have shown an excess of TF motif turnover within regulatory DNA (see work from Duncan Odom's group, Vierstra el., 2014 Science for a direct mouse-human comparison).

We appreciate Dr. Vierstra’s insights into the relationship between TF spacing and evolutionary conservation of TF binding sites. We have included discussion about this in the Discussion section. Specifically, the lack of requirement for exact spacing and remarkable tolerance of spacing alterations by TFs with relaxed spacing could potentially associate with the high turnover of TF binding sites found by previous studies (Vierstra et al., 2014), although further investigation would be needed to establish this association.

3. While Leveraging CRISR/Cas9 editing to generate a broad spectrum of 'spacing' alleles is a clever approach to tackle the experimentally test the effect of motif spacing on TF (co-)binding, the experiment is very underpowered to test the generalizability of the authors claims. Did the authors select single cell clones from the editing experiment or just look at bulk edited populations? If the former, it is unclear how any conclusions can be made from a mixture of edited cells that have a spectrum of indels (also likely carrying two different alleles).

We thank Dr. Vierstra for pointing out that the description of the analysis method of InDels was not clear. To perform these experiments, we prepared bulk populations of cells following CRISPR mutagenesis. Importantly, the ChIP-seq libraries were prepared by selective amplification of ChIP tags containing the targeted region of interest. Thus, for each region-specific sequence tag that was immunoprecipitated, we could simultaneously determine whether an InDel had been created and its specific length. Each tag is thus cell- and allele-specific. Since there was a mixture of edited cells with a spectrum of InDels, we first estimated the distribution of InDels based on the tags of input DNAs and then compared it with TF ChIPseq tags to associate TF binding with different sizes of InDels, in which the effect of an InDel is reported as the odds ratio of ChIP tags to the input tags. By focusing on specific loci, extremely deep tag counts were achieved (703686 to 16189879 tags/locus), allowing strong statistical conclusions. We revised the methods diagram in Figure 5A and further clarified our approach in the revised manuscript.

Reviewer #3:Transcription factors (TFs) bind to the DNA in a sequence-specific manner at TF binding sites (TFBSs) to control gene transcription. Hence, characterizing how TFs interact with DNA is key to uncover how gene regulation occurs and how this process can be disrupted in diseases. While the binding properties of a large portion of human TFs are well characterized, a remaining challenge lies in our knowledge of how TFs interact cooperatively at regulatory elements, either forming dimers or co-binding the same regions. In this manuscript, Shen et al. explored spacing patterns between TFBSs. Relying on ChIP-seq data, they developed a new methodology to predict TF pairs harbouring constrained or relaxed spacing patterns between their TFBSs. The authors made their code available, which allows reproducibility and exploration; this should be a requirement in the field but is not always complied with so we thank the authors for this. When applied to a limited set of TFs with ChIP-seq data in K562 cells, the authors predicted that TF pairs primarily bind to DNA with relaxed spacing between their TFBSs. Nevertheless, they were able to highlight already known as well as novel specific pairs of TF harboring constrained spacing. Next, the authors leveraged naturally occurring small insertions and deletions in the human population and mouse strains to confirm that altering spacing between TFBSs of TF pairs with relaxed spacing patterns has limited effect. This observation was further supported by synthetic spacing alterations induced by CRISPR-Cas9 experiments. The study is overall well designed and addresses an important challenge in our understanding of TF-DNA interactions and TF cooperation.Nevertheless, we believe that there are some methodological limitations that favor the identification of relaxed spacing patterns, which should be better outlined in the manuscript to allow the reader to fully comprehend the results. From the title and first sections of the manuscript the readers are given the impression that relaxed and constrained spacing instances are about to be described and analysed with an equal importance. However, more focus is given to the relaxed spacing with both the mouse and CRISPR analyses exclusively dedicated to this with no clear explanation why. It would be useful to the readers to have this explicitly outlined by the authors. Finally, the terminology associated with TF-DNA interactions is very often incorrect, which confuses the readers and should be addressed. Please see below for our detailed comments.

We thank Reviewer #3 for the positive comments and constructive suggestions. In response to the comment on the relative consideration of relaxed vs constrained spacing instances, the emphasis on relaxed instances was motivated by extensive prior work on constrained spacing instances. We modified the text so that readers are not given the impression that relaxed and constrained instances will be considered equally.

1. The terminology associated with TF binding events is inappropriate. The authors use "ChIP-seq peaks", "TFBSs", and "motifs" almost interchangeably, which is not correct. The inconsistency in the terminology makes it difficult to fully comprehend what the authors meant/did.An example is one of the first sentences in the Introduction: "TFs bind to short, degenerate sequences at promoters and enhancers, often referred to as TF binding motifs." The sequences bound by TFs in promoters and enhancers are TFBSs while TF binding motifs are computational representations of TFBS sets, which can be represented in many ways such as consensus motifs, PFMs, etc.The next sentence claims that "TFs bind in an inter-dependent manner to closely spaced motifs." Motifs cannot be closely spaced but TFBSs are. Another example is the subsection "Motif identification" in the Methods section while the authors describe the prediction of TFBSs (using motifs).

We appreciate reviewer’s insights into our usage of these terminologies. We have made sure in our revision that these terminologies are used in a consistent way. Specifically, “motifs” were used to describe an aggregation of DNA sequences recognized by TFs, which can be represented as PWMs. “TF binding sites” were used for individual sequences matching with motifs, and “ChIP-seq peaks” for enriched regions in ChIP-seq data originally identified using HOMER.

2. More details should be provided to the Methods section. We acknowledge that the authors provide their code for inspection but outlining all methodological details in the manuscript would help in the clear understanding of the methods used. For instance:a. The authors do not describe how they selected de novo motifs using HOMER (only best motif?, any p-value threshold?, any specific background used?)

We have provided more details under “TF binding site identification” section in Methods.

b. For the TFBS predictions, the authors used a FPR threshold of 0.1%, but which was the specific tool that they used for that? FPR computation depends on background expectation, what was used (e.g. 25% A, C, G, or T or nucleotide composition of the genome)?

We have added the tool and background expectation for FPR computation under “TF binding site identification” section in Methods.

c. P. 23, lines 466-470. The authors described that they conducted a permutation test but then described that the null distribution was obtained using random spacing values between 0 and 100. If the null distribution is obtained by randomly selecting values between 0 and 100, it does not correspond to a permutation. A permutation test would imply permuting the observed spacing values.

We have added the tool and background expectation for FPR computation under “TF binding site identification” section in Methods.

d. P. 25, lines 513-516. It is not clear to us why the authors considered subsets of mutations when overlapping TFBSs predicted for the ChIP'ed TFs (only if 2-bit difference in motif score) but not for mutations overlapping TFBSs predicted by MAGGIE (all mutations). Why not consider all mutations in all cases?

We did require motif score difference to be at least 2-bit to be considered as a motif mutation for both ChIP’ed TFs and MAGGIE-predicted TFs during our analyses. To clarify our procedure, we included more details in Methods under the “Categorization of genetic variation based on impacts on motif or spacing” section. The threshold of 2-bit was set to reduce false positives of motif mutations, which could have a small sequence alteration but likely do not affect TF binding.

3. The methodology used has several limitations that are not described by the authors. We encourage the authors to clearly outline them to the readers. Furthermore, we believe that these limitations favor the identification of relaxed spacing, which should be acknowledged, especially since the majority of the work focuses on alterations of spacing for TF pairs with relaxed spacing patterns.a. It is well documented that TF binding preferences for TFs binding in close proximity (e.g. as dimers) can be altered. For instance, Jolma et al. (https://www.nature.com/articles/nature15518) used CAP-SELEX to reveal that "Most TF pair sites identified involved a large overlap between individual TF recognition motifs, and resulted in recognition of composite sites that were markedly different from the individual TF's motifs." As the authors relied on TF binding profiles (or motifs) corresponding to the binding preferences of TFs binding as monomers, it is possible that they will miss cooperative binding inducing a change in binding preferences. Furthermore, the authors did not consider overlapping motifs, which is again precluding the identification of constrained spacing.

Our work is distinguished from Jolma et al. by focusing on TFs with independent motifs at nonoverlapping positions, and because of that, we potentially missed TFs that should be regarded as a complex and recognize composite motifs all the time, which took up the majority of the 315 TF-TF interactions discovered by Jolma et al. We discussed this as a limitation in our original submission and have expanded the original discussion to clarify the difference of our work from Jolma et al. in the Discussion section.

b. The authors rely on ChIP-seq data to identify TF cooperation. While this is fine overall, this data does not allow the authors to know whether two identified TFs bind to their TFBSs on the same molecules. Indeed, ChIP-seq being a bulk experiment, it does not allow to discriminate between true co-occupancy on the same molecule or not. This should be discussed to put the work in a larger context to the readers. See https://www.cell.com/molecular-cell/pdf/S1097-2765(20)30793-0.pdf for a reference.

We agree with this comment and have included a discussion referencing Sönmezer et al. in the Discussion section.

c. The de novo motif enrichment does not ensure that the motif found is actually the one bound by the ChIP'ed TF. Indeed, the motifs found for ARID1B and ARID2 correspond to GATA motifs while ARID TFs bind to more general A+T rich motifs. It is unclear whether the signal observed for these TFs is due to their direct interaction with DNA.

The de novo motifs used for identifying TF binding sites were manually selected from the top three significant HOMER motifs that looks like motifs of other TFs within the same TF family available in JASPAR. We added these details of our motif selection under “TF binding site identification” section in Methods. Meanwhile, we appreciate reviewer for pointing out that the motifs we used for ARID1B and ARID2, despite being A+T rich, are very similar to GATA motifs, so we excluded those two TFs in our revision to reduce confusion.

4. It would have been nice to perform similar experiments as the mouse and CRISPR ones but considering TF pairs with fixed spacing of their TFBSs. If the authors decide not to, they should at least clearly state to the readers that they more specifically focused on relaxed spacing and why.

As noted in our general response, we focused on instances of relaxed spacing for CRISPR mutagenesis because instances of fixed spacing have been investigated previously. We added a clear statement saying our focus on relaxed spacing for the mutagenesis analyses of InDels at the end of section “Natural genetic variants altering spacing between relaxed transcription factors are associated with less deleteriousness in human populations” in the Results section. As a summary of our reasons for focusing on relaxed spacing, our work together with several other studies found that relaxed spacing was more commonly seen among collaborative TFs, but very few studies have discussed the effects of altered spacing on relaxed TFs. We note that we extended the analysis presented in the previous submission for CEBPB by also carrying out ChIP-seq for SPI1 or PU.1 in the CRISPR-edited cells. These studies fully reproduced the patterns observed for CEBPB and are provided in Figure 5—figure supplement 2.

5. It would have been interesting to provide information about the prevalence of cobinding events in the different ChIP-seq datasets. For instance, what is the percentage of ChIP-seq peaks that contain a predicted TFBS for each TF? What is the percentage of such peaks that contain cobinding events? Is there a difference in numbers b/w constrained or relaxed spacing. Providing this information would help the readers to put these observations into a more general context of TF binding regions.

The numbers and percentages of ChIP-seq peaks containing predicted TF binding sites have been included in the new Figure 1-table supplement 2. The number of peaks with co-binding events were originally shown in Figure 1B, and we made a new Figure 1B to show percentages in parallel. There were only marginal differences between constrained and relaxed spacing according to a two-sample t-test (p=0.03), so we decided not to include the results in the revision, but reviewer can find the distributions below.

6. It would be nice to have similar plots as in Figure 3B for pairs of TFs with constrained spacing to show how it contrasts.

As reviewer suggested, we have included the plots as new Figure 1—figure supplement 5 and discussed these new results in the section “Transcription factors primarily co-bind with relaxed spacing” of the Results section.

7. In all analyses, it seems that more deletions than insertions were observed (see Figure 2A for instance). It would be interesting to see if the results recapitulate when considering insertions and deletions independently.

We performed separate analyses for insertions and deletions as reviewer suggested. Results were consistent between insertions and deletions and were included as a new Figure 2—figure supplement 2 in our revision.

8. It is unclear to us what is the analysis of MAGGIE-predicted TFBSs adding to the story. Moreover, the authors claim that MAGGIE identifies "functional" TFBSs but the authors do not provide any specific evidence of function (and which function?) in our opinion.

We thank Reviewer 3 for raising this point. MAGGIE is used to identify motifs of transcription factors that serve as collaborative binding partners for the reference transcription factor being analyzed by ChIP. For example, if mutations in the recognition motif for TF-A are significantly associated with reduced nearby binding of TF-B as determined by ChIP, we conclude that TF-A is a collaborative binding partner of TF-B. This relationship then serves as the basis for analysis of spacing relationships between TF-A and TF-B in the absence of direct binding data for TF-A. In this sense, MAGGIE identifies ‘functional’ motifs because they are associated with loss of binding of the reference TF. We clarify this in the revised manuscript.

9. It seems that there is a periodic pattern in Figure 5E with log-odds ratio periodically equal to 0 (at least with indel sizes < -45). Could this correspond to the minor groove width periodicity of ~10bp (see for instance https://www.cell.com/cell/pdf/S0092-8674(18)31312-6.pdf for periodicity of mutations)? Could the authors comment on that?

Those large deletions <-45 bp disrupt motifs directly, so it would be hard to tell if the “periodic pattern” in Figure 5E reflects DNA periodicity. If there is any periodicity, it should appear in those deletions altering spacing as well. Therefore, we think that further investigation would be needed to understand those patterns.

10. In several figures, the authors should provide all points instead of summarizing with boxplots, which are frowned upon as they hide data distribution.

As reviewer suggested, all data points have been added to the boxplots in Figure 3C,E,G, Figure 4C, and Figure 5C.

11. P. 9 line 178: the authors mentioned 50 millions SNPs but we do not find where this data is used as observing SNPs would not alter spacing.

SNPs were used when we looked for motif mutations. We clarified accordingly in the Methods section under “Categorization of genetic variation based on impacts on motif or spacing”.

12. P. 2 line 46: "implicating their effect…" we would rather write "suggesting their effect…".

We changed the texts as reviewer suggested.

13. The authors seem to have ignored homodimers. Maybe their methodology could be extended to consider spacing b/w TFBSs for the same TF.

Many motifs in JASPAR database are already based on homodimers or heterodimers (e.g., p65, C/EBP, AP-1). We therefore considered all such factors as recognizing a single motif. If we interpret the comment regarding spacing between TF binding sites for the same TF, we agree that this would be of interest. However, such sites are difficult to identify as objectively as sites where two different TFs bind, because the ChIP-peaks for the same TF would overlap if collaborative binding interactions occurred within the spacing ranges.

14. TF naming is sometimes inconsistent in the text and figures. For example, SPI1 and PU.1 are both used in Figure 3. Similarly, we recommend revisiting the text for p65 and RELA, as well as C/EBPβ and CEBPB.

To maintain consistency as reviewer suggested, we changed SPI1 to PU.1, CEBPB to C/EBPβ, and RELA to p65 when referring to transcription factors in Figure 3 and supplementary figures.

15. It is not clear where the authors retrieved HepG2 cell line data from (used in the Figure 1 —figure supplement 4). Was this data processed in the same way as K562 cell line data?

We added the source and processing description of HepG2 data in the Results section.

16. P. 8 line 152: we suggest replacing "we overlaid" with "we mapped".

We changed the texts as reviewer suggested.

17. P. 1 line 30: "ChIP-sequencing" should be replaced with "ChIP-seq" for consistency.

We changed the texts as reviewer suggested.

18. Figure 3D,F,H : instead of motif score this should be TFBS score.

We changed “motif score” to “PWM score” in our texts to align with the most commonly used terminology in literature.